# Latent Feature Mining for Predictive Model Enhancement with Large Language Models

## Abstract

Predictive modeling often faces challenges due to limited data availability and quality, especially in domains where collected features are weakly correlated with outcomes and where additional feature collection is constrained by ethical or practical difficulties. Traditional machine learning (ML) models struggle to incorporate unobserved yet critical factors. In this work, we introduce an effective approach to formulate latent feature mining as text-to-text propositional logical reasoning. We propose *FLAME* (**F**aithful **L**atent Fe**A**ture **M**ining for Predictive **M**odel **E**nhancement), a framework that leverages large language models (LLMs) to augment observed features with latent features and enhance the predictive power of ML models in downstream tasks. Our framework is generalizable across various domains with necessary domain-specific adaptation, as it is designed to incorporate contextual information unique to each area, ensuring effective transfer to different areas facing similar data availability challenges. We validate our framework with two case studies: (1) the criminal justice system, a domain characterized by limited and ethically challenging data collection; (2) the healthcare domain, where patient privacy concerns and the complexity of medical data limit comprehensive feature collection. Our results show that inferred latent features align well with ground truth labels and significantly enhance the downstream classifier.

## 1 Introduction

Prediction plays a crucial role in decision-making across many domains. While traditional machine learning (ML) models are powerful, they are often constrained by the availability of observed data features. Contrary to the common belief that we are in a "big data era," this is not always the case, especially in areas where decisions have profound impacts on human lives. In areas like criminal justice and healthcare, data availability is often constrained, with ethical limitations further restricting the features that can be collected and used (Lu et al., 2021; Yuan et al., 2023). As a result, many critical decisions must rely on a limited set of features, some of which may have weak correlations with the prediction target. This presents significant challenges for achieving accurate predictions.

To overcome the challenges posed by limited feature availability and quality, latent feature mining is a common approach. However, traditional techniques face two key limitations in domain-specific applications. First, inferring domain-specific latent features often requires contextual information beyond the available data, such as expert input, public information, or crowd-sourcing. This information is typically in natural language, which ML models like neural networks struggle to process and encode into proper embeddings. Second, many latent feature mining techniques, such as deep-learning based auto-encoders and the Expectation-Maximization (EM) algorithm, lack interpretability. They extract features in abstract mathematical formats that are difficult to explain in human terms. This is especially problematic in high-stakes domains like healthcare or criminal justice, where explaining and justifying a model's predictions is crucial for building trust and ensuring ethical decision-making. The black-box nature of these methods makes it harder to gain confidence in the model's outputs in these domains.

Figure 1 illustrates the motivation behind our approach to address these two limitations. Human experts can infer additional latent features that go beyond the explicit data provided by drawing on their experience. For example, in the criminal justice system, predicting an individual's likelihood

The predictive model performance is **terrible** … We only have limited number of features …

Based on my **domain knowledge**, I know some **crucial latent features** might help with the prediction ! And I can infer these features from given features using my expertise !

But how do I scale my ability to infer latent features ? Can LLMs mimic my inference process?

Figure 1: The real-world example illustrating the motivation of *FLAME*, a framework to augment observed features collected in given datasets with latent features.

of in-program recidivism (the probability of committing a new crime during probation) is crucial for determining eligibility for incarceration-diversion programs (Rotter & Barber-Rioja, 2015; Li et al., 2024). Typically, available data includes only basic demographic and criminal history information, but domain knowledge suggests that other factors – such as socio-economic status, community support, and psychological profiles – can significantly impact outcomes. Collecting such sensitive data raises ethical concerns, but human case managers can rely on their professional experience to infer these critical yet unrecorded details from observed data. While effective, this human-based approach is difficult to scale, as it relies on tacit human knowledge that is hard to formalize into standardized processes. Additionally, the human reasoning process is both time- and labor-intensive, limiting its application to large populations.

Recent advancements in large language models (LLMs) present a promising new avenue with their advanced reasoning capability (Brown et al., 2020; Ouyang et al., 2022; Achiam et al., 2023). LLMs have potential to process and generate information in ways that mimic human thought processes (Ji et al., 2024). Building on this insight, we propose *FLAME*, a framework that leverages LLMs to augment observed features with latent features and enhance the predictive power of ML models in downstream tasks like classification. *FLAME* offers two key advantages over traditional latent feature mining methods: (1) it seamlessly integrates contextual information provided in natural language, and (2) by emulating human reasoning, it produces more interpretable outputs, making it particularly valuable in high-stakes domains requiring explainability. We summarize our main contributions as follows.

1. We introduce a new approach that LLMs to formulate latent feature mining as a reasoning task using text-to-text propositional logic. This method effectively infers latent features from observed data and provides significant improvements in downstream prediction accuracy and interpretability over traditional techniques.

2. We develop a four-step versatile framework that integrates domain-specific contextual information with minimal customization efforts. This framework is highly adaptable across various domains, particularly those with limited observed features and ethical constraints on data collection.

3. We empirically validate our framework through case studies in both the criminal justice and healthcare sectors, where latent features play an important role in enhancing prediction tasks. The framework's strong performance in two different application settings demonstrates its adaptability and usefulness for other domains facing similar challenges.

## 2 RELATED WORKS

**Data Augmentation versus Latent Feature Mining** Data augmentation is a technique widely employed to provide more data samples to improve the predictive power of ML models (Van Dyk & Meng, 2001). Generative models such as Generative Adversarial Networks (GANs) learn data patterns and generate synthetic data to augment training sample sizes (Goodfellow et al., 2014; Kingma & Welling, 2013). In contrast, latent features are hidden characteristics in a dataset that are not directly observed but can be inferred from available data. Incorporating meaningful latent features can enhance the performance of downstream applications (Zhai & Peng, 2016; Jiang et al., 2023). Methods such EM and Variational Autoencoders (VAEs) offer alternative techniques to infer latent features from observed data. However, EM algorithms, while estimating latent variable assignments and updating model parameters to maximize data likelihood, often produce results that are difficult to interpret and require strong parametric assumptions. Similarly, VAEs use probabilistic approaches

to describe data distribution with latent variables, but the learned mappings can also be hard to interpret. Another related approach is dimension reduction such as Principal Component Analysis, which reduces the size of the feature space while preserving the most important information. However, dimension reduction is less effective when the input feature set is already limited.

We summarize a comparison in table 2 to further distinguish the difference between *FLAME* and existing approaches for enhancing predictive model from data/features perspective.

| Methods | Approach | Interpretability | Contextual Information Integration Capability |
|---|---|---|---|
| Data Augmentation (GANs, VAEs) | increasing sample size | ✕ | ✕ |
| Latent Feature Mining (EM) | extracting (new) latent features | ✕ | ✕ |
| Dimension Reduction | reducing feature size | ✕ | ✕ |
| FLAME | extracting (new) latent features | ✓ | ✓ |

Table 1: Comparison of *FLAME* and related methods: Unlike data augmentation, which increases sample size, *FLAME* expands the feature space by training LLMs to infer latent variables from existing features. Compared to traditional latent feature mining methods, *FLAME* mimics human expert reasoning and incorporates domain-specific context, offering improved interpretability. Unlike dimension reduction methods, *FLAME* enriches the dataset by adding latent features that capture key aspects of the underlying phenomena.

**Fine-tuning for LLMs Training.** Fine-tuning is an effective method for LLMs to reduce hallucinations and better align outputs with real-world data and human preferences (Tonmoy et al., 2024; Qiao et al., 2022; Hu et al., 2021). Synthetic data offers a low-cost way to enhance LLM reasoning across domains (Liu et al., 2024; Zelikman et al., 2022; Wang et al., 2022). *FLAME* also uses synthetic data during fine-tuning, but unlike prior work that directly mimics observed features, we are among the first to treat synthetic latent feature generation as a reasoning task. Through few-shot prompting, *FLAME* creates synthetic "rationales" for the reasoning process to infer latent features, followed by fine-tuning to enhance accuracy and reduce hallucinations.

Note that we distinguish between augmenting the feature space and augmenting training data. Our primary goal is to enrich the feature space by inferring and adding latent features to improve downstream predictions. As part of the steps in *FLAME* to achieve this goal, we also augment training data with synthetic samples during the fine-tuning process for LLMs.

## 3  THE PROBLEM SETTING

In this section we formally describe our problem setting that leverages latent features to enhance downstream tasks. The downstream task we focus on is a multi-class classification problem, but the framework can easily extend to other downstream prediction tasks such as regression problems.

**Definition of Latent Features.**
Latent features, denoted as $Z$, represent underlying attributes that are not directly observed within the dataset but are correlated with both the observed features $X$ and the class labels $Y$. We use a function $g$ with $Z = g(X)$ to denote the correlations between the latent features and the observed features $X$. As shown in figure 3, latent features $Z$ are correlated with $X$ and $Y$. One can learn the latent features from the original features $X$ and augment the features $f(\mathbf{X}, \mathbf{Z})$ to learn the classifier $Y$.

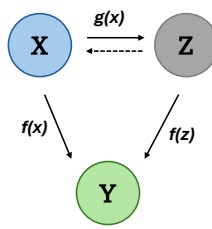

In a standard multi-class classification problem setting, suppose we have a dataset $D = (x_1, y_1), (x_2, y_2), \ldots, (x_n, y_n)$, where $x_i$ is a $d$-dimensional vector representing the input features $X \in \mathcal{X}$ and $y_i \in \mathcal{Y} = \{1, 2, \ldots, C\}$ denotes the corresponding class label $Y$ for individual $i = 1, \ldots, n$. The goal is to learn a classifier $f : \mathcal{X} \to \mathcal{Y}$ that accurately predicts the class labels. Consider the following scenarios in which $f$ struggles to capture the relationship between $X$ and $Y$: (1) The number of input features $X$ is small relative to the complexity of the classification

task. (2) When $X$ are weakly correlated with class labels $Y$, they may not provide discriminating information to accurately predict the corresponding class labels.

To address these challenges, we can use additional informative features to enhance the classifier's ability to capture the relationship between $X$ and $Y$. Latent features can serve such a purpose (See ***Definition of Latent Features*** in Page 3).

While this approach seems beneficial intuitively, it is important to note that adding more features is not always helpful if the extracted features are not meaningful and introduce noise. In the following lemma, we show in a simple logistic regression setting that while adding features can reduce in-sample loss, it does not always reduce out-of-sample loss if the added features are not informative. We use the log-loss (the cross-entropy loss) of the logistics regression for binary outcome $Y \in \{0, 1\}$. We denote the optimal coefficients that minimize the in-sample log-loss function as $\beta^*$ for the original features and $\tilde{\beta}^*$ for the augmented features.

**Lemma 1.** *The in-sample log-loss always follows $\mathcal{L}^{in}(\tilde{D}, \tilde{\beta}^*) \leq \mathcal{L}^{in}(D, \beta^*)$. When the added features are non-informative, there exist instances such that the out-of-sample log-loss $\mathcal{L}^{out}(\tilde{D}, \tilde{\beta}^*) > \mathcal{L}^{out}(D, \beta^*)$.*

The results in the lemma can be generalized to multi-class labels. Since augmenting the feature space is not necessarily beneficial unless the added features are meaningful, a major part of our case study is to empirically test whether the extracted features from our framework indeed improve downstream prediction. If the added features significantly enhance downstream prediction accuracy, this provides strong evidence that the inferred latent features are meaningful.

## 4 LATENT FEATURE MINING WITH LLMS

We propose a new approach, *FLAME*, to efficiently and accurately extract latent features and augment observed features to enhance the downstream prediction accuracy. It extracts the latent features $Z$ from the original features $X$ to capture complex patterns and relationships that individual features may overlook, especially when some of the $X$'s are weakly correlated with the outcome $Y$. At a high level, our approach transform this latent feature extraction process as a text-to-text propositional reasoning task, i.e., infer the relationship $Z = g(X)$ through logical reasoning with natural language. Figure 2 provides an example of the extract process with the steps elaborated on below.

Following the framework established in previous work (Zhang et al., 2022), we denote the predicates related to the observed features as $P_1, P_2, \ldots, P_m$. Consider a propositional theory $S$ that contains rules that connect $P$'s to the latent feature $Z$. We say $Z$ can be deduced from $S$ if the logic implication $(P_1 \wedge P_2 \wedge \ldots \wedge P_m) \to Z$ is covered in $S$. For potentially complicated logical connections between $P$'s and $Z$, we also introduce intermediate predicates $O$'s and formulate a logical chain (a sequence of logical implications) that connects $X$ to the latent features $Z$ as follows:

$$X \to (P_1 \wedge P_2 \wedge \ldots \wedge P_m) \to (O_1 \wedge O_2 \wedge \ldots \wedge O_\ell) \to Z. \tag{1}$$

Our approach formulates this logical chain as a multi-stage Chain of Thoughts (CoT) prompt template, and then guide LLMs to infer $Z$ from $X$ using the prompt template. Specifically, we first extract predicates $P$'s from $X$. Then we infer intermediate predicates with a rule $(P_1 \wedge P_2 \wedge \ldots \wedge P_m) \to O_l$ for $l = 1, \ldots, \ell - 1$, and forward the intermediate predicates into the next stage to infer $O_{l+1}$. Finally, we infer latent features with $(O_1 \wedge O_2 \wedge \ldots \wedge O_\ell) \to Z$. With the formulated multi-stage CoT prompt template, we then generate synthetic training data to fine-tune LLMs to enhance the logical reasoning ability of LLMs in the self-instruct manner (Wang et al., 2022).

We use a hypothetical example from our case study setting to illustrate the formulation of the logic chain. The blue (leftmost) box in Figure 2 shows the observed feature $X$ for one individual. Examples for the predicates $P$'s formulated from $X$ could be:

> $P_1$ :*"the client has part-time job"*, $P_2$ : *" the client hasn't complete high school"*,
> $P_3$ :*"the client is single"*, $P_4$ : *"the client has drug issue"*, $P_5$ :*" the client lives
> in high crime area"*, $P_6$ : *" the client is assessed with high risk"* ...

To infer the latent feature $Z$ – in this example, the support likely needed during probation – we go through a multi-stage reasoning to infer the intermediate predicates $O$'s; see the white (middle) boxes in Figure 2. One example logic that connects $P$'s to $O$'s could be:

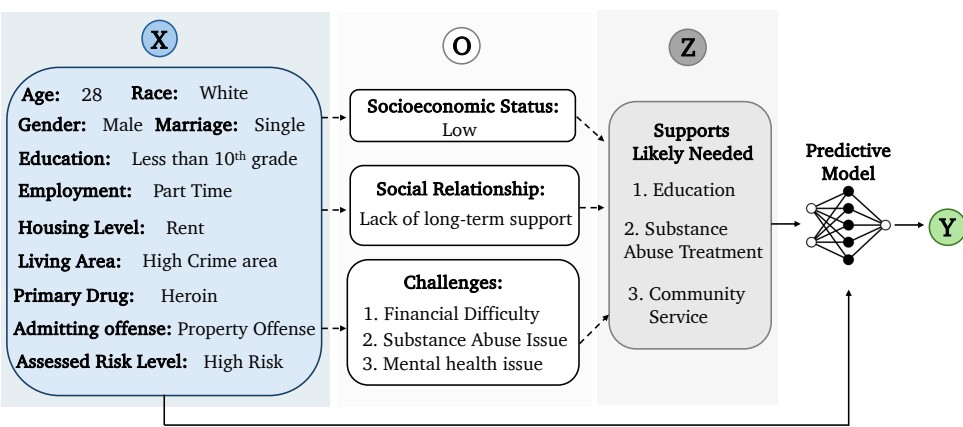

Figure 2: Example of latent feature mining through chain of reasoning. The latent feature "Supports Likely Needed" ($Z$) is inferred from the observed input features ($X$) via intermediate predicates ($O$), and is then used alongside $X$ to improve the prediction for outcome ($Y$).

$P_1$ = "The client has unstable employment"
$P_2$ = "The highest education level of client is less than 10th grade"
$O_1$ = "The client has low socioeconomic status"
If $(P_1 \wedge P_2 \rightarrow O_1) \in S$, then $O_1$ is True.

Finally, with $P$'s and $O$'s, we can connect $X$ with $Z$ though the logic chains. One example of the logical chain is as follows:

> "The client is grappling with unstable employment and a relatively low educational level, factors that likely contribute to a low socioeconomic status. Additionally, being single, struggling with drug issues, and residing in a high-crime area further exacerbate the lack of positive social support. Given these circumstances, education could be valuable. Community service can be particularly beneficial for someone who is single and may lack a broad support network. Substance abuse treatment is crucial for individuals from lower socioeconomic backgrounds to aid in recovery from substance abuse. Hence this client likely needs support on education, substance abuse treatment, community service."

Here, *"unstable employment and a relatively low educational level"* and *"being single, struggling with drug issues, and residing in a high-crime area"* are $P$'s extracted from the features $X$, while *"a low socioeconomic status"* and *"lack of positive social support"* are $O$'s. Finally, the rationales *"education could be valuable . . . recovery from substance abuse. Hence this client likely needs support on education, substance abuse treatment, community service"* connect the intermediate predicates to the latent variables $Z$ (supports likely needed) we want to infer, i.e., $Z_1$='education', $Z_2$='substance abuse treatment', $Z_3$='community service'.

Figure 3 illustrates the full process of of *FLAME* with four steps.

**(1) Formulate baseline rationales:** The first step is to formulate baseline rationales, whic serve as guidelines for LLMs to infer latent features from observed ones. This involves two sub-steps:

– The first sub-step is to develop some baseline rationales, i.e., identify observed features potentially correlated with latent features and formulate their relationships – the logic chain that connects $X$ to $Z$. Sources to help formulate these baseline rationales include established correlations (e.g., risk score formulas), expert input, and other contextual information like socio-economic status in the neighborhood. This is also a critical step in our framework that allows the **integration of domain-specific contextual information** in the format of natural language.

– In the second sub-step, we craft prompts with interactive alignment. This is a critical component to establish correct reasoning steps for prompts used in Step 2 to generate synthetic rationales. We involve experienced human in the domain to provide a prompt template for LLMs to generate

rationales aligned with the baseline rationales, then test the prompt template on a few examples using zero-shot. If the LLM fails to certain example, we provide the ground truth back to the LLM, allowing it to revise the prompt template (Miao et al., 2023). This process iteratively refines the template until LLMs consistently generate the desired output for all selected examples.

**(2) Enlarge data with synthetic rationales for fine-tuning:** We generate synthetic training data in self-instruct fashion (Wang et al., 2022). With a handful of examples of the baseline rationales as a reference, we guide the LLMs via in-context learning to generate similar rationales to enlarge the training data samples. To ensure the quality and diversity of the generated dataset, we introduce human-in-the-loop interventions to filter out low-quality or invalid data based on heuristics. We also leverage automatic evaluation metrics for quality control, e.g., removing data that lack essential keywords.

**(3) Fine-tuning LLMs:** To enhance the reasoning capabilities of the LLMs and better align their outputs in specific domains, we leverage the fine-tuning process with processed dataset from the previous step (Qiao et al., 2022). Fine-tuning not only boosts the accuracy and reliability of the LLMs, but also significantly improves their ability to reason with complex inputs and reduce hallucination (Tonmoy et al., 2024).

**(4) Latent feature inference:** The fine-tuned model mirrors the nuanced reasoning process of human experts. We use it to infer latent features, which are then fed into downstream prediction tasks to improve accuracy.

## 5 EXPERIMENTS SETUP

We design two case studies to empirically investigate the following questions: (1) Can *FLAME* accurately mimic human reasoning to infer latent features? (2) When labels for latent features are available, is *FLAME* more effective than conventional methods in predicting the labels? (3) Does *FLAME* improve the performance of downstream prediction tasks?

### 5.1 CASE STUDY 1: INCARCERATION DIVERSION PROGRAM MANAGEMENT

In this case study, we conduct evaluation of *FLAME* on a unique dataset from a state-wide incarceration diversion program as described in Appendix F. Specifically, We designed two tasks to answer the three questions. Task (1) Risk Level Prediction (Section 5.1.1): we treat the risk level of individuals as a latent feature, despite it being collected in the dataset (i.e., true labels are available). This experiment examines whether the latent features $\hat{Z}$ inferred by LLMs match well with the actual features $Z$. (2) Outcome Prediction (Section 5.1.2): we assume that the "supports likely needed" are latent features, which lack ground truth labels. We first have LLMs infer these features, then add them to the downstream prediction task of program outcomes $Y \sim f(X, \hat{Z})$ and evaluate whether the prediction accuracy is improved. That is, the inferred features are indeed beneficial and not detrimental (recall the results in Lemma 1).

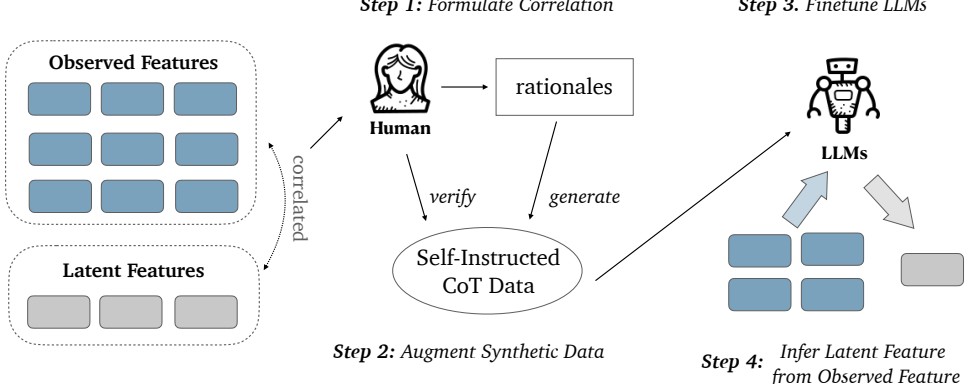

Figure 3: Overview of latent feature inference framework.

### 5.1.1 Risk Level Prediction

**Task Description.** In this task we treat an observed feature –Risk Level – as the latent feature to infer. The task is a multi-classification problem to learn $Z \sim g(X)$ among four labels for the latent variable $Z \in \{moderate, high, very\_high\}$ based on each client's profile $X$.

**Implementation Details.** We implement our proposed framework as follows. All prompt templates are available in Appendix C.

- Step 0. Profile writing: In this pre-processing step, we translate structured data $X$ into text that can be better handled by LLMs, i.e., formulating predicates $P$'s from the features $X$. Then we formulate the intermediate predicates $O$'s, where we prompt LLMs to extract and summarize underlying information such as background, socio-economic status, and challenges in two or three sentences. We then merge these sentences into the client's profile. We use zero-shot prompting with GPT-4.

- Step 1. Formulating rationales: Using human input, established risk score calculations (Corrections), and the code book with risk calculation details provided by our community partner, we summarize a general rule for inferring risk levels from the profiles, i.e., establishing the logic chains from $P$'s and $O$'s to $Z$. We sample 40 client features from the dataset and formulate 40 baseline rationales that logically connect features to corresponding risk levels and are aligned with the high-level general rule. To avoid the primacy effect of LLMs, we rate risk scores from 0 to 10 to add variability in the labels, categorized as follows: 0-4 (moderate risk), 4-7.5 (high risk), and 7.5-10 (very high risk).

- Step 2. Enlarge fine-tuning data: With the 40 baseline rationales, we generate additional synthetic rationales. We sample client features and corresponding ground truth risk scores from the dataset, using one of the 40 rationales as an example, to prompt LLMs to produce similar narratives with CoT prompts. In total we got 3000 rationales for the training data.

- Step 3. Fine-tune LLMs: Our framework is designed to be plug-and-play, allowing the synthetic data generated in the previous step to be used across different language models. We fine-tune two pre-trained language models for cross-validation purposes: GPT-3.5 and Llama2-13b (OpenAI, 2021). We use OpenAI API to fine-tune GPT-3.5-turbo-0125 (Touvron et al., 2023; OpenAI). We fine-tune Llama2-13b-chat using LoRA (Hu et al., 2021).

- Step 4. Inference with LLMs: We prompt fine-tuned LLMs to infer risk level $\hat{Z}_i$ from features $X_i$ for each client $i$ in the test data and evaluate the out-of-sample accuracy by comparing the inferred latent variable (risk level) $\hat{Z}_i$ with the ground truth label $Z_i$.

**Evaluation.** We choose ML classifiers (e.g., Neural Networks) as the baseline to directly predict $\hat{Z}_i$ from features $X_i$ using the given class labels. We compare the prediction performance of $\hat{Z}_i$ inferred from *FLAME* with that from ML models using out-of-sample accuracy and F1 score. Additionally, we evaluate the quality of generated text with an automatic evaluation metric. In the pre-processing step, we assess the keyword coverage rate in the generated profile assuming each feature value is a keyword. For synthetic rationales, we use YAKE, a pretrained keyword extractor (Campos et al., 2020), to identify keywords, and then evaluate the keyword coverage rate with a rule-based detector to determine how many logical information points are covered.

### 5.1.2 Outcome Prediction

**Task Description.** In this task, we treat the "support likely needed" (e.g., substance treatment, counseling) for each client as the latent features $Z$ and use them to augment the original feature $X$ for outcome prediction, which is a multi-classification problem to learn $Y \sim f(X, Z)$ among four labels for the outcome $Y \in \{Completed, Revoked, NotCompleted, Other\}$. The raw dataset does not record this feature, thus, $Z$ in this task is truly unobservable (in contrast to the one used in the first task). Available support program options for this task are detailed in Appendix F.3.

**Implementation Details.** Steps 0 and 2-4 remain almost the same as in the risk-level prediction task. Step 1 requires a slight adjustment (as discussed in Section 4, this step is the main part in our framework that requires customization). Here, we formulate 40 baseline rationales in step 1 to deduce "support likely needed" from client features. We leverage multi-stage prompting strategy (Qiao et al., 2022) to break down the task into three sub-tasks: (1) identify the main challenges

from the client's profile, (2) rank these challenges by priority, (3) match the challenges with suitable programs. Particularly, the third task is our main goal, with the first two serving as steps to streamline the process and simplified the task.

**Evaluation.** We train an ML classifier to predict outcomes with and without the inferred latent features, i.e., $\hat{Y}_i \sim f(X_i, \hat{Z}_i)$ versus $\hat{Y}_i \sim f(X_i)$. We evaluate the out-of-sample accuracy by comparing the predicted outcome $\hat{Y}_i$ with the true label $Y_i$ in the test data. This comparison allows us to assess whether incorporating the latent features enhances the classifier's performance.

## 5.2 CASE STUDY 2: HEALTHCARE MANAGEMENT

In this case study, we test the efficacy of *FLAME* in the healthcare domain. We conduct experiments on MIMIC dataset (Johnson et al., 2016), a comprehensive dataset containing detailed de-identified patient clinical data (see more in Appendix F).

**Task Description.** The discharge location prediction task involves using individual patient-level data to predict the most likely discharge destination for patients upon their discharge from the hospital inpatient units. We apply *FLAME* to extract (new) latent features to enhance the prediction accuracy for this discharge location task. Specifically, we create a new feature, "social support," which captures the extent of healthcare, familial, and community support available to the patient after being discharged.

**Implementation Details.** We repeat the four-step process of our framework[1]: Steps 0 and 2-4 remain almost the same as in the previous two tasks. We leverage domain expertise to help us craft rationales to infer social support in Step 1.

**Evaluation.** Same as Section 5.1.2, we train an ML classifier to predict outcomes with and without the inferred latent features, i.e., $\hat{Y}_i \sim f(X_i, \hat{Z}_i)$ versus $\hat{Y}_i \sim f(X_i)$ and then evaluate their out-of-sample accuracy.

## 6 EXPERIMENTS RESULTS

In this section, we demonstrates the experiment results. We also conduct ablation experiments to further investigate our advantage and limitations (Please see Appendix D).

### 6.1 RISK LEVEL PREDICTION RESULTS

**Generated Text Quality.** For profile writing in Step 0, we treat each individual feature in $X_i$ as a keyword to cover, and measure the keyword coverage rate. The generated profiles demonstrated an average keyword coverage rate of 98%. For the generated synthetic rationales in Step 2, we treat terms such as age, gender, employment, and education as critical keywords and assess their coverage rate. The fine-tuned GPT-3.5 and Llama2-13b-chat both achieved a keyword coverage rate of 100%. This indicates that the generated content adheres strictly to the guidelines established in the training data, ensuring that all necessary information is accurately represented.

**Latent Variable Inference Performance.** As shown in Figure 4(a), our approach achieves the highest overall accuracy. The reason that ML models struggle to predict well is due to the fact that there is no strong correlation between the observed features and the targets (risk level); see the correlation plot in Appendix F.4. In contrast, our approach demonstrates superior performance, since it more effectively handles datasets with subtle or non-obvious relationships between the observed and target variables. This result shows that **our approach is able to make accurate inference of latent features and outperforms traditional ML approaches**.

Table 4(b) details the prediction performance by class, showing F1 scores for each class using ML models and our approach. Notably, all ML models struggle with the 'Very High Risk' category – this category is often misclassified as 'High Risk' due to similar feature distributions of these two categories and unbalanced data (only 371 training points for 'Very High Risk'). In contrast, our approach significantly improves the prediction performance for this category, highlighting its **effectiveness for unbalanced datasets.** This improvement is likely because our LLM-based approach

---

[1]We released the code of the implementation for reproducing and evaluation. Please access the code through the anonymous link: https://bit.ly/3XMi8QN

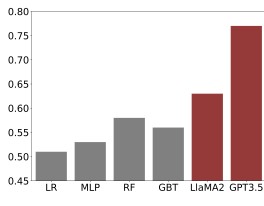

(a) Model accuracy

| Category | LR | MLP | RF | GBT | LLaMA2 | GPT3.5 |
|----------|-----|-----|-----|-----|--------|--------|
| Moderate | 51% | 54% | 44% | 46% | 57% | **69%** |
| High | 65% | 55% | 69% | 66% | 70% | **81%** |
| Very High | 20% | 11% | 18% | 18% | 38% | **81%** |

(b) F1 scores

Figure 4: Risk level prediction results: (a) Model accuracy; (b) F1 scores per-category. LR - logistic regression; MLP - Neural Networks; RF- random forest; GBT - Gradient Boosting Trees.

has intermediate steps (profile writing to obtain the socio-economic status and other contextual factors in step 0 and connecting these factors with the latent variables in step 1), which help capturing the subtle distinctions between 'High Risk' and 'Very High Risk' that are not explicitly recorded.

## 6.2 OUTCOME PREDICTION RESULTS

We compare the performance of the downstream classifiers that trained with and without the latent features. Note that in the first task (risk-level inferrence), GPT3.5 demonstrated better performance than llama2-13b. Thus, we focused on fine-tuning GPT-3.5 when using our approach for this task.

| without latent feature | LR | MLP | GBT |
|------------------------|----|-----|-----|
| ROC AUC Score (std.) | 70% (0.01) | 81% (0.01) | 84% (0.01) |
| F1 Score (std.) | 70% (0.01) | 70% (0.01) | 71% (0.01) |

| with latent feature | LR | MLP | GBT |
|---------------------|----|-----|-----|
| ROC AUC Score (std.) | **85% (0.02)** | **88% (0.01)** | **92% (0.01)** |
| F1 Score (std.) | 75% (0.01) | 73% (0.01) | 77% (0.01) |

(a) Model Performance

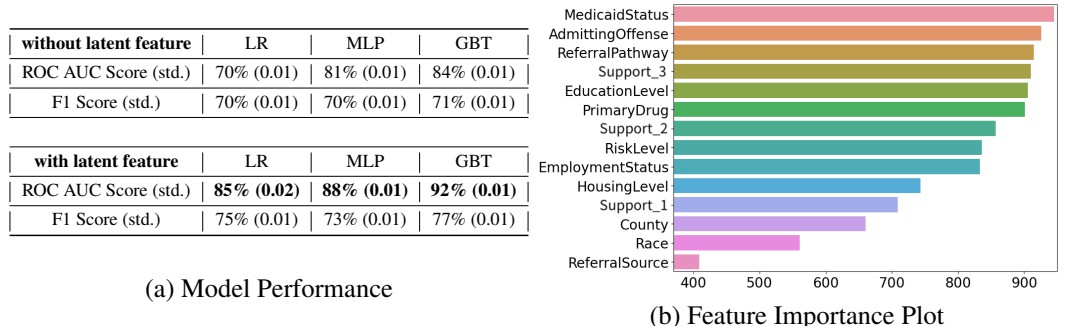

(b) Feature Importance Plot

Figure 5: Outcome prediction results: (a) Model performance with/without the inferred latent features (program requirements); (b) feature importance plot. LR - logistic regression; MLP - Neural Networks; GBT - Gradient Boosting Trees.

As illustrated in Table 5(a), incorporating latent features significantly improves the performance of the downstream classifiers. Furthermore, the feature importance in Figure 5(b) shows that the inferred features – 'Support_1', 'Support_2', and 'Support_3' – are among the top-ranked features. This implies the significant relevance of these features on the downstream classification task. Hence, we can conclude that **our approach has the capability of enhancing the downstream classifier's accuracy with inferred latent features**.

## 6.3 DISCHARGE LOCATION PREDICTION RESULTS

| Model | Accuracy (std.) | F1 score (std.) |
|-------|-----------------|-----------------|
| LR | 65.22% (0.01) | 65.46% (0.01) |
| MLP | 63.19% (0.02) | 63.19% (0.02) |
| GBT | 64.84% (0.01) | 65.09% (0.01) |
| RF | 65.11% (0.01) | 65.44% (0.01) |
| **LR w/ Latent Feature** | **71.22% (0.01)** | **71.26% (0.01)** |
| **MLP w/ Latent Feature** | **74.40% (0.01)** | **74.50% (0.01)** |
| **GBT w/ Latent Feature** | **75.56% (0.02)** | **75.38% (0.02)** |
| **RF w/ Latent Feature** | **75.31% (0.01)** | **75.22% (0.01)** |

Table 2: The experiment result for Discharge Location Prediction task. We use five different random seeds to run experiment five times and report the average.

Table 2 demonstrates the result of discharge location prediction task. The results show an average improvement of approximately 8.64% in accuracy and 8.64% in F1 score when latent features are added to the models. This is similar to the percentage increase reported in Table 5(a). Specifically, the GBT model achieves the highest accuracy after incorporating the latent features. The results demonstrate another strong evidence of using our framework to improve downstream prediction power with the addition of latent features.

Furthermore, as shown in Figure 12 in the appendix, the inferred variable "Social Support" shows strong correlation with the discharge location. This finding suggests that *FLAME* can uncover meaningful latent variables that might otherwise be overlooked in traditional data collection methods in the healthcare settings. More importantly, this experiment on a different dataset from a different domain demonstrates the effectiveness and generalizability of *FLAME*.

## 7 DISCUSSION AND CONCLUSION

In conclusion, *FLAME* provides a novel solution to the challenges of limited feature availability in high-stakes domains by using LLMs to augment observed data with interpretable latent features. This framework improves downstream prediction accuracy while enhancing explainability, which makes it valuable for sensitive decision-making in areas like healthcare and criminal justice.

**What is required to generalize *FLAME* for each new application?**   *FLAME* has broad potential across various domains, particularly those with limited observed features and ethical constraints. Steps 2-4 primarily rely on the adaptability of LLMs and allow flexible application across different domains. However, Step 1 – identifying and formulating baseline domain-specific rationales – requires domain expertise and involves additional manual effort. This effort is worthwhile because our framework is intentionally designed to be domain-specific. We believe this is actually the critical step that drives the improved downstream prediction accuracy demonstrated in Section 6. By leveraging contextual information that traditional methods cannot, *FLAME* significantly enhances model performance.

To elaborate, in Step 1, we utilize contextual information to tailor the framework to the specific domain. For example, in the outcome prediction task (Section 5.1.2), we incorporated external public information on the socio-economic status of different zip codes. Our ablation study showed that excluding this zip code information significantly reduced the LLM's ability to extract useful latent features, which highlights the importance of this contextual data in enhancing predictive power. Moreover, Step 1 allows human to provide external contextual information to align the LLM's reasoning and to mitigate potential issues raised from the LLM's inherent knowledge limitations. In another ablation study, we prompted GPT-4 to directly generate contextual information for zip codes based solely on its internal knowledge, without external input. Out of 50 zip codes, 5 could not be determined due to lack of information, 17 provided incorrect (hallucinated) information, and only 33 were correct (see Appendix E for examples). This result is consistent with recent research findings that LLMs are not reliable as knowledge bases (He et al., 2024; Zheng et al., 2024).This shows that, although our method requires more manual effort than other ML-based latent feature mining methods, it effectively integrates contextual information that traditional approaches cannot, which makes it both more effective in mining domain-specific features and worth the investment.

**Future work.**   As we continue to refine our *FLAME* framework, we are actively pursuing avenues to enhance its fidelity and reliability. First, we are streamlining the process to reduce the need for human intervention and increase the scalability of our approach. Second, we acknowledge that LLMs can inadvertently perpetuate existing biases present in their training data, and how to mitigate such bias remains an open question in the field Wan et al. (2023); Gallegos et al. (2024). *FLAME* attempts to minimize such biases by leveraging domain-specific data and expert input during the fine-tuning process. Furthermore, the training dataset is curated to include a diverse range of scenarios, and the model's inferences are continually tested against ground truth data where available. Nevertheless, we are implementing more sophisticated error control mechanisms to diminish the impact of potential inaccuracies in the generated features. For example, we are in the process of hiring human annotators to verify the output from the LLMs reasoning. Other possible options include developing confidence scoring systems for generated features (Detommaso et al., 2024).

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

APPENDIX

## A    PROOF OF LEMMA 1

We use the log-loss, defined as

$$\mathcal{L}(D, \beta) = -\frac{1}{n} \sum_{i=1}^{n} [y_i \log(p_i) + (1 - y_i) \log(1 - p_i)] \tag{2}$$

for given data $D = \{(x_i, y_i)\}_{i=1}^{n}$ and $p_i = 1/(1 + e^{-(\beta_0 + \beta_1 x_i)})$. When using the augmented feature $\tilde{x}_i = (x_i, z_i)$, we denote the data as $\tilde{D} = \{((x_i, z_i), y_i)\}_{i=1}^{n}$.

For the first part of the lemma, we note that the in-sample log-loss for the original features follows

$$\mathcal{L}^{\text{in}}(D, \beta) = -\frac{1}{n} \sum_{i=1}^{n} [y_i \log(p_i) + (1 - y_i) \log(1 - p_i)], \tag{3}$$

and the in-sample log-loss for the augmented features follows

$$\mathcal{L}^{\text{in}}(\tilde{D}, \beta) = -\frac{1}{n} \sum_{i=1}^{n} [y_i \log(\tilde{p}_i) + (1 - y_i) \log(1 - \tilde{p}_i)], \tag{4}$$

where $p_i = 1/(1 + e^{-(\beta_0 + \beta_1 x_i)})$ and $\tilde{p}_i = 1/(1 + e^{-(\beta_0 + \beta_1 x_i + \beta_2 z_i)})$.

We denote the optimal coefficients that minimize the log-loss in equation 3 as $\beta^* = (\beta_0^*, \beta_1^*)$, and the coefficients that minimize the log-loss in equation 4 as $\tilde{\beta}^* = (\tilde{\beta}_0^*, \tilde{\beta}_1^*, \tilde{\beta}_2^*)$. Note that $\check{\beta} = (\beta_0^*, \beta_1^*, 0)$ is a feasible solution for the log-loss in equation 4. Therefore, using the optimization property, we have

$$\mathcal{L}^{\text{in}}(\tilde{D}, \tilde{\beta}^*) \leq \mathcal{L}^{\text{in}}(\tilde{D}, \check{\beta}) = \mathcal{L}^{\text{in}}(D, \beta^*),$$

which completes the first part of the lemma.

For the second part of the lemma, we first assume that for the given data $D$, $\mathcal{L}^{\text{in}}(\tilde{D}, \tilde{\beta}^*) = \mathcal{L}^{\text{in}}(D, \beta^*) - \epsilon/n$ where $\epsilon \geq 0$ from the first part of the lemma. We now construct an instance with an out-of-sample dataset $D'$ that contains $n + 1$ samples, where $D'$ consists of (i) the $n$ data points that exactly match with $D$ (or $\tilde{D}$) for the first $n$ samples, and (ii) one additional sample $(x_{i+1}, y_{i+1})$ (or $((x_{i+1}, z_{i+1}), y_{i+1})$ when using the augmented features). Without loss of generality, assume that $y_{i+1} = 1$. Then we have

$$\mathcal{L}^{\text{out}}(D', \beta^*) = \frac{1}{n+1} \left( n\mathcal{L}^{\text{in}}(D, \beta^*) - \log(p_{i+1}) \right)$$

and

$$\mathcal{L}^{\text{out}}(\tilde{D}', \tilde{\beta}^*) = \frac{1}{n+1} \left( n\mathcal{L}^{\text{in}}(\tilde{D}, \tilde{\beta}^*) - \log(\tilde{p}_{i+1}) \right).$$

When the added features $Z$'s are non-informative, we consider the scenarios that they are noise and the additional term $\tilde{\beta}_2^* Z$ also contributes noise to the predictions. In other words, the coefficients $\tilde{\beta}^*$ do not generalize well to the test data. Therefore, there exists an instance where the realization of $Z$, $z_{i+1}$ deviates from the predicted probability significantly, such that

$$\tilde{p}_{i+1} < p_{i+1}/\exp(\epsilon) \leq p_{i+1}.$$

Note that this instance exists since the noise terms do not correspond to any actual pattern in the test data, causing incorrect predictions, and in our construction, a smaller predicted probability would be less accurate as the label $y_{i+1} = 1$. Therefore,

$$-\log(\tilde{p}_{i+1}) > -\log(p_{i+1}) + \epsilon,$$

and

$$\begin{aligned}
\mathcal{L}^{\text{out}}(\tilde{D}', \tilde{\beta}^*) &= \frac{1}{n+1} \left( n\mathcal{L}^{\text{in}}(D, \beta^*) - \epsilon - \log(\tilde{p}_{i+1}) \right) \\
&> \frac{1}{n+1} \left( n\mathcal{L}^{\text{in}}(D, \beta^*) - \log(p_{i+1}) \right) = \mathcal{L}^{\text{out}}(D', \beta^*).
\end{aligned}$$

## B  COMPUTE RESOURCES

For all experiments, we split data into training and testing dataset with ratio of 8:2.

For experiment 1 (risk level prediction), we finetune LLaMA2-13b-chat on 2 X NVIDIA RTX A6000 for 4 hours with LoRA. And we finetuned three times for different subtasks. We use OpenAI offical API to finetune GPT3.5 model, which requires no GPUs. Each finetune job takes about 2 hours. We repeat 3 times for different sub tasks. Additionally, we also run Machine Learning baseline model on CPU (Intel i7). We run grid search for each classifier.

For experiment 2 (outcome prediction), we use OpenAI offical API to finetune GPT3.5 model, which requires no GPUs. Each finetune job takes about 2 hours. We repeat 6 times for different sub tasks.Additionally, we also run Machine Learning baseline model on CPU (Intel i7). We run grid search for each classifier.

All other experiments (e.g. sensitive experiment) are conducted on ChatGPT, which requires no GPU.

## C  PROMPT TEMPLATE

```
Task: Write a paragraph to profile the client, please include following:

1. Write sentences to cover all basic information provided.
2. Provide information about the area of this client live in, as much more details as you can.
3. Infer social economic status of this client
4. Infer the challenges that this client might facing.

Here are the basic information of the client: <features>.

Here is the reference of living area context: <additional info>
```

Figure 6: Profile writing prompt

```
Here is the profile of a client: <profile>
Given the client's information, please infer a risk score out of 10.

Given client's information to infer risk score out of 10, we know that:
1. Employment (If client has unstable employment status, increase the score by 1.
Adjust score if needed): ___
2. Financial Status (If client has financial difficulty, increase the risk score by 1.
If client relies on social economic assistance, further increase the risk score by 1.
Adjust score if needed.): ___
3. Education (Increase the risk score by 1 if the highest grade of school completed is
less than grade 12. Further increase the risk score by 1 if the highest grade completed
is less than grade 10): ___
4. Family and Marital (Increase score if client is dissatisfied with his/her current
marital relationships situation. Increase risk score if the client is a social isolate.
Adjust score if needed.): ___
5. Drug (Increase risk score by 1 if the client has ever had a drug problem. If the
drug problem is related with Heroin, further increase the risk score by 1. Adjust score
if needed.): ___
6. Living Area (Increase risk score by 1 if the client lives in a high crime
neighborhood): ___
7. Age (Increase risk score by 0.3 if the client is under the age of sixteen): ___
8. Gender (Increase risk score by 0.3 if the client is male): ___
Conclusion: ___
```

Figure 7: Risk Level Prediction: Prompt template and response CoT template

```
Here is the profile of a client: <profile>
Analyze the provided profile of the client to infer the main challenges he faces.
```

```
Given the identified challenges for the client, infer the priority of each
challenge in terms of immediate action and long-term impact on his reintegration
into society. Please response in the ranking order. Here are the challenges: Here
are the challenges <challenges>:
```

```
Here is the available list of programs <program list>:
Given the profile and challenges of the client, select the top 3 program
requirements that would be most beneficial for the client.
Here is the profile of client: <profile + top 3 ordered challenges>
```

Figure 8: Requirement selection: Multi-stage Prompt template

```
To select the top 3 programs that would be most beneficial for the client, let's analyze each
available options:
1. Thinking for a Change (It aims to transform criminogenic thinking patterns with designed
cognitive-behavioral curriculum. Recommend for clients assessed at relatively high risk
level): __
2. Employment (It aims to help client develop employability. Recommend this for clients with
unstable employment status): __
3. Education (It aims to engage clients in educational programs. Recommend clients without a
high school diploma or GED):__
4. Positive Peer Mentoring (It offers positive role models and fosters a supportive network,
which can deter criminal associations. Recommend this for clients residing in high-crime
areas):__
5. Community Service (It aids in building a sense of responsibility and community connection.
Recommend for clients with property offense or drug-related offenses):__
6. Mental Health Treatment (It addresses underlying mental health issues that may contribute
to criminal behavior. Recommend for clients with a history of substance abuse or unstable
living conditions):__
7. Anger Management (It focuses on teaching effective emotion and reaction management
techniques. Recommend for clients who exhibit aggressive behaviors or have property-related
offenses):__
8. Substance Abuse Treatment (It aims to help clients overcome substance dependencies.
Recommend for clients with histories of drug-related offenses or primary drug use):__
9. Domestic Violence Counseling (It aims to address and modify violent behavior patterns.
Recommend for clients involved in violent incidents):__
10. Sex Offender Counseling (It focuses on behavior modification and preventing recidivism.
Recommend for clients with sex-related offenses):__
Conclusion: ___
```

Figure 9: Requirement selection: Response CoT template

# D  ABLATION STUDY

**Do the inherent biases of LLMs influence the inference process of latent features?**  To assess whether the reasoning processes within generated texts exhibit biases, we conducted the following experiments. First, we utilized the pretrained keyword extraction model YAKE (Campos et al., 2020) to search for racial terms within the reasoning steps of the generated text. The analysis showed that such keywords were absent, indicating no explicit racial bias in this context. Second, we closely examined the race distribution in the ground-truth data versus the distribution in the predictions made by the model. The analysis revealed that the race distributions between the ground-truth and the predicted outcomes were similar. This similarity suggests that the model does not introduce additional racial biases in its predictions and accurately reflects the distributions present in the input data. Both results validate that the LLMs' inherent biases are not carried into the inference process. Other types of bias, such as bias in lexical context, are beyond the scope of this paper and are left for future research.

**How sensitive is our approach to the quality of human guidelines?**  *FLAME* is sensitive to human guidelines, specifically the baseline rationales and prompt templates formulated in Step 1. We have conducted an ablation study to determine the optimal level of details required in the prompts. As shown in Figure 10 (b), the best performance was achieved with the most reasoning steps and a sentence length of two per step. In other words, increasing the number of reasoning steps allows us to decompose the task into simpler components and enhances the performance of LLMs. More importantly, while human guidelines are important, **the interactive self-revise alignment strategy can significantly help** during the sub-step of Step 1 (prompt crafting). By providing ground truth and encouraging self-reflection, GPT-4 can revise the prompt template to include crucial details, ensuring a more accurate evaluation.

**How important is the fine-tuning step in *FLAME*?**  We have conducted another ablation study, where we repeated the risk-level prediction task with zero-shot, one-shot, and three-shot prompting to compare with our fine-tuned model. In zero-shot, we provided only the task description. In one-shot and three-shot, we included randomly selected human-verified examples. Accuracy rankings from lowest to highest were: three-shot (40%), zero-shot (55%), one-shot (60%), and the fine-tuned model (75%); see Figure 10 (a). The three-shot's poor performance may be due to information loss from long inputs. Zero-shot responses are highly variable and not well-suited for downstream tasks. Although one-shot showed improvement, the fine-tuned model significantly outperformed all others. Hence, the answer to the question is that **fine-tuning is necessary**. Additionally, the fine-tuning process incorporates feedback loops with domain experts to adjust and correct the model's reasoning pathways, ensuring that the latent features inferred, such as the need for substance abuse treatment, are aligned with nuanced real-world outcomes rather than broad statistical correlations.

| Setting | Accuracy |
|---------|----------|
| Zero-shot | 55% |
| One-shot | 60% |
| Three-shot | 40% |
| Fine-tune | 75% |

(a) Risk level prediction results across different setting

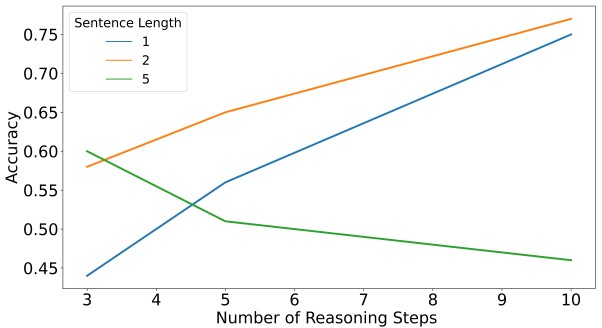

(b) Risk level prediction results across different strategy

Figure 10: Ablation study results: (a) Experiments on risk level prediction task using GPT4 with different prompting setting. (b) Experiments using GPT4 with different prompting setting different prompting strategies.

**How does `FLAME` perform compared to the baseline?**    We implemented a baseline approach using an MLP classifier on embeddings from a text summarization model, and repeat the risk level prediction experiment: We use few-shot to generate a profile that contains all information related to the client ( See Figure 6 for more detail on prompt template ), then we extract the embedding from the encoder of Pegasus (Zhang et al., 2019). These embeddings serve as input to an MLP classifier for risk level prediction. We evaluated on a balanced validation set of 50 samples per class, and this baseline achieved only 52% accuracy on the three-class classification task, while our proposed approach is able to reach over 75% accuracy.

# E    EXAMPLE OF CONTEXTUAL INFORMATION EXTRACTION EXPERIMENT

In this section, we demonstrate three example of responses: correct response with correct extracted contextual information, incorrect response with hallucinated false contextual information, and incorrect response with no extracted contextual information.

For each example, we used the prompt "What is the socioeconomic status for people live in the area around *zipcode* in 2022? Please response from median income, poverty rate, education level, and employment rate aspect." We validate the result with the zipcode tabulation area data from census.gov [2]

## E.1    EXAMPLE OF CORRECT RESPONSE

**Zipcode**    02108

**Response**    The area around ZIP code 02108 corresponds to parts of downtown Boston, Massachusetts. As of 2022, here are some socio-economic statistics typically associated with this area, based on available data:1. Median Income: The median household income in Boston's downtown area, including 02108, is generally higher than the national average. In 2022, the median household income in Boston was approximately $73,000, but localities within the 02108 ZIP code, particularly those closer to affluent neighborhoods and business districts, may experience median incomes significantly above this figure, potentially exceeding $100,000.2. Poverty Rate: The poverty rate in Boston, and particularly in the downtown areas, is lower than the national average. In 2022, the city's overall poverty rate was around 17%, but this can vary significantly within different neighborhoods. The downtown area, including 02108, tends to have a lower poverty rate, likely ranging below 10%.3. Education Level: The education level in this region is generally high. Boston is home to many universities and colleges, which contributes to a highly educated population. In 2022, about 50% of the residents in the city had a bachelor's degree or higher, with the figure likely being higher in the downtown area due to the presence of professionals and students. 4. Employment Rate: The employment rate in Boston was strong in 2022, with unemployment rates below the national average, around 3-4%. The economic activities in downtown Boston support a vibrant job market, particularly in finance, healthcare, and education. For precise statistics and potential updates, it's advisable to refer to local government resources or community organizations that provide detailed demographic and economic reports.

## E.2    EXAMPLE OF HALLUCINATED RESPONSE

**Zipcode**    46034

**Response**    The area around ZIP code 46034 corresponds to Cicero, Indiana. As of 2022, here are some socio-economic statistics typically associated with this area, based on available data: 1. Median Income: The median household income in Cicero was remarkably high at $200,000 in 2022, driven by the area's booming tech startups and an influx of affluent professionals attracted by luxury housing developments. 2. Poverty Rate: The poverty rate was exceptionally low, reported at just 0.5%, thanks to the area's robust economic growth and comprehensive local government initiatives that offer extensive support services for education, housing, and employment. 3. Education Level: Nearly 90% of the adult population held advanced degrees (master's or higher), making Cicero one

---

[2]Sample zipcode data from census.gov: https://data.census.gov/profile/16875?g=860XX00US16875

of the areas with the highest densities of PhD holders in the country, many of whom work remotely for prestigious universities and research facilities. 4. Employment Rate: Employment in the 46034 area was at an all-time high, with 99.9% of the population gainfully employed in 2022, reflecting the area's vibrant and highly competitive job market. For precise statistics and potential updates, it's advisable to refer to local government resources or community organizations that provide detailed demographic and economic reports.

### E.3 EXAMPLE OF RESPONSE WITH LACK OF ACCURATE INFORMATION

**Zipcode**   99501

**Response**   As of my last knowledge update in October 2023, I do not have access to real-time databases for the most current socioeconomic statistics, including those specifically for the area around the ZIP code 99501 in 2022. However, I can provide general information about how you might find this data and what typical sources you could check for the most accurate and relevant information:1. Median Income: The U.S. Census Bureau is a primary source for median income data. The American Community Survey (ACS) provides detailed income statistics, which you could look up specifically for the 99501 ZIP code.2. Poverty Rate: The poverty rate can also be found in the ACS data. This survey includes information on the number of people living below the poverty line.3. Education Level: Data on education levels (including high school graduation rates, college degrees, etc.) is also available through the ACS. This information can give insight into the educational attainment of the residents in the area.4. Employment Rate: Employment statistics, including unemployment rates and employment by various sectors, can be reviewed through the Bureau of Labor Statistics (BLS) and the ACS.For the most accurate and specific information regarding the socioeconomic status of the area in and around ZIP code 99501, I recommend checking the following resources:- U.S. Census Bureau: Visit their official website and look for the American Community Survey, where you can enter the ZIP code for detailed demographic information.- Local Government Websites: City or state government websites may also provide socioeconomic profiles of their respective areas.- Local Libraries or Economic Development Agencies: These organizations often keep datasets on the local economy and demographics.If you need detailed statistics for that specific area, accessing these resources should provide you with the necessary information.

## F   DATASET BACKGROUND

### F.1   INCARCERATION-DIVERSION PROGRAMS AND DATA DESCRIPTION

Incarceration-diversion programs aim to support individuals who have committed minor offenses by providing community-based services to improve societal reintegration and reduce recidivism. Eligible individuals were diverted from traditional incarceration to such programs after risk assessment and screening. Case managers determined specific program requirements, such as substance use treatment and cognitive-behavioral therapy. There are four types of program outcomes: Completed (successfully completed the program), Revoked (committed new crimes while in the program), Not Completed (unable to finish for various reasons), and Other (unrecorded reasons).

We obtained de-identified data from our community partner for a state-wide incarceration-diversion program in Illinois. The consolidated dataset includes records of adult participants admitted to the program. The collected data features include timestamps such as the arrival and termination dates to the program, program outcomes, and individual features such as the race, gender, education, county, marriage status, housing, risk assessment scores, prior crime history, and sources of referral (e.g., from probation officer or from the court).

## F.2 INCARCERATION DIVERSION DATA DESCRIPTION

Table 3: Categorical Covariates Summary Statistics (N/A or Other Categories are Omitted).

| Variable | Categories | County | | | |
|---|---|---|---|---|---|
| | | DuPage | Cook | Will | Peoria |
| Risk | Highest | 24.3 | 32.0 | 2.3 | 1.0 |
| | High | 60.7 | 26.2 | 35.1 | 24.7 |
| | Medium | 11.0 | 15.6 | 42.1 | 47.0 |
| AdOffense | Drugs | 43.0 | 67.8 | 31.7 | 37.0 |
| | Property | 31.1 | 17.6 | 52.5 | 46.3 |
| | DUI | 11.1 | 2.3 | 3.8 | 1.0 |
| OffenseClass | Class 4 | 42.5 | – | 11.5 | 20.6 |
| | Class 3 | 13.5 | – | 5.7 | 5.7 |
| | Class 2 | 16.0 | – | 5.7 | 5.1 |
| Pdrug | Heroin | 27.0 | 43.6 | 32.3 | 9.5 |
| | THC | 18.6 | 18.5 | 17.5 | 21.6 |
| | Coc.Crack | 7.8 | 10.9 | 21.0 | 11.6 |
| ReferralReason | Tech Violation | 31.2 | 0.0 | 12.8 | 0.0 |
| | 3/4 Felon | 20.5 | 70.5 | 59.2 | 80.0 |
| | 1/2 Felon | 9.8 | 16.5 | 23.7 | 14.7 |
| WhoReferred | Prob Officer | 64.7 | 97.3 | 1.8 | 0.0 |
| | Judge | 32.0 | 1.3 | 0.7 | 91.3 |
| | Pub. Defender | 0.6 | 0.0 | 75.3 | 2.8 |
| Gender | Female | 25.2 | 21.3 | 21.7 | 19.8 |
| | Male | 74.8 | 77.5 | 78.2 | 80.0 |
| EmplymntS | Full Time | 49.7 | 85.7 | 38.2 | 6.7 |
| | None | 32.3 | 4.8 | 59.2 | 92.0 |
| | Part Time | 18.0 | 9.4 | 2.7 | 1.3 |
| MaritalS | Single | 86.4 | 85.6 | 15.0 | 22.9 |
| | Married | 5.9 | 7.1 | 1.8 | 5.7 |
| | Divorced | 4.7 | 2.3 | 0.2 | 1.8 |
| EducationS | HighSchool | 40.3 | 37.2 | 34.3 | 13.6 |
| | No HighSchool | 32.6 | 52.4 | 10.8 | 12.3 |
| | Some College or Graduated | 19.4 | 3.5 | 11.8 | 4.4 |
| HousingS | Friend or Family | 62.3 | 27.9 | 6.2 | 17.7 |
| | Own/Rent | 29.0 | 15.5 | 2.7 | 11.1 |
| | No Home Reported | 5.9 | 23.9 | 16.5 | 70.2 |
| MedicaidS | Yes | 23.8 | 48.4 | 8.3 | 3.3 |
| UniqueAgents | 4 | 11.6 | 2.2 | 8.6 | – |
| | 3 | 27.9 | 31.9 | 22.3 | 2.3 |
| | 2 | 60.6 | 65.9 | 69.1 | 97.7 |
| FinalProgPhase | Level 3/4 | 11.1 | 15.7 | 32.3 | 0.3 |
| | Level 1/2 | 56.5 | 14.4 | 22.7 | 3.1 |
| | Level 0 | 2.9 | 35.5 | 7.0 | 27.0 |
| RewardedBehv | Yes | 4.0 | 29.1 | 2.5 | 1.5 |
| Sanctions | Yes | 91.8 | 99.3 | 89.8 | 41.1 |

### F.3  INCARCERATION DIVERSION PROVIDED SUPPORTS

| Support Name | Description |
|---|---|
| Thinking for a Change | Aimed at transforming criminogenic thinking patterns using a cognitive-behavioral curriculum, recommended for clients at a high risk level. |
| Employment | Helps develop employability, recommended for clients with unstable employment status. |
| Education | Engages clients in educational programs, recommended for those without a high school diploma or GED. |
| Positive Peer Mentoring | Provides positive role models and a supportive network, recommended for clients in high-crime areas. |
| Community Service | Builds a sense of responsibility and community connection, recommended for clients with property or drug-related offenses. |
| Mental Health Treatment | Addresses underlying mental health issues, recommended for clients with a history of substance abuse or unstable living conditions. |
| Anger Management | Teaches emotion and reaction management techniques, recommended for clients who exhibit aggressive behaviors or have property-related offenses. |
| Substance Abuse Treatment | Helps overcome substance dependencies, recommended for clients with drug-related offenses or primary drug use. |
| Domestic Violence Counseling | Addresses and modifies violent behavior patterns, recommended for clients involved in violent incidents. |
| Sex Offender Counseling | Focuses on behavior modification and preventing recidivism, recommended for clients with sex-related offenses. |

Table 4: Available Supports

## F.4 FEATURE CORRELATION MATRIX FOR INCARCERATION DIVERSION DATA

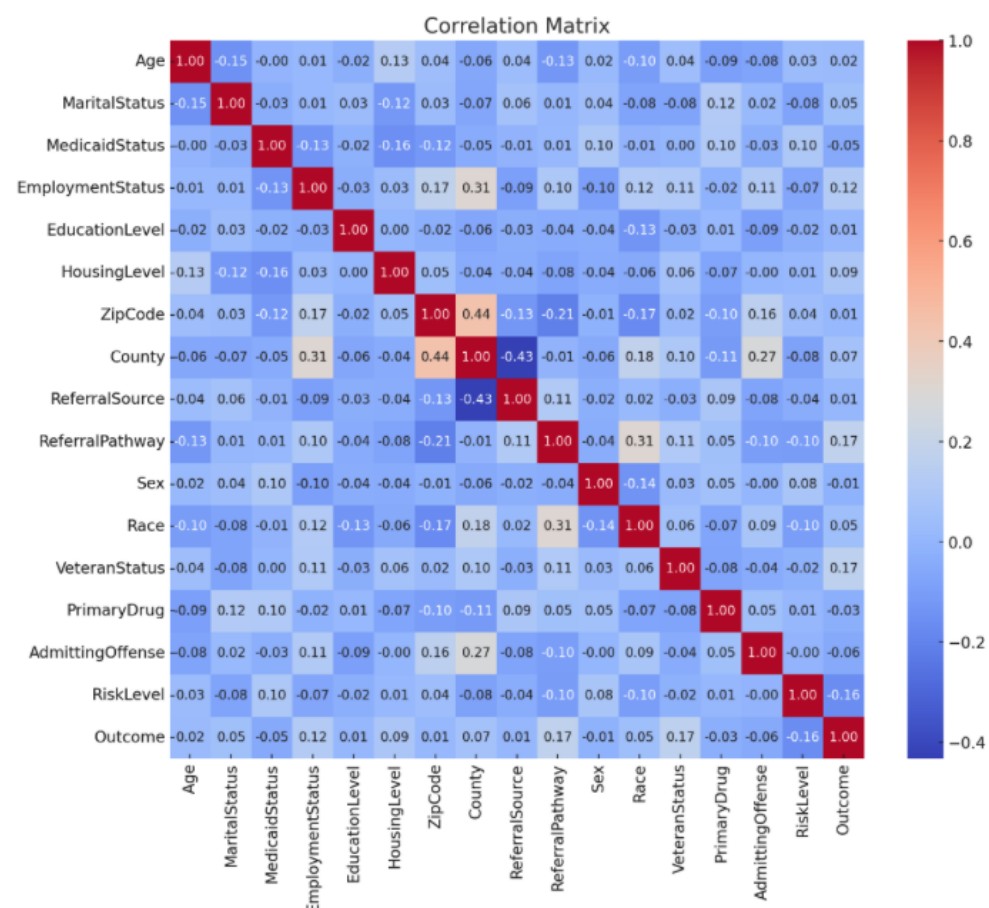

Figure 11: Correlation Matrix of Features for Incarceration Diversion Data

### F.5 ELECTRONIC HEALTH RECORD DATA DESCRIPTION

MIMIC (Medical Information Mart for Intensive Care) dataset is a comprehensive dataset containing detailed de-identified patient clinical data and is widely used for various prediction tasks in the machine learning literature.

### F.6 ELECTRONIC HEALTH RECORD DATA DESCRIPTION

Table 5: Categorical Variables Summary Statistics

| Variable | Categories | Percentage |
|---|---|---|
| Discharge Location | Home | 40.19 |
| | Other | 40.19 |
| | Died | 19.62 |
| Gender | Female | 51.53 |
| | Male | 48.47 |
| Race | White | 61.09 |
| | Black/African American | 11.70 |
| | Other | 11.45 |
| | Asian | 2.49 |
| | Hispanic or Latino | 1.89 |
| | White - Other European | 1.69 |
| Marital Status | Married | 43.05 |
| | Single | 35.29 |
| | Widowed | 11.01 |
| | Other | 10.65 |
| Insurance | Other | 58.24 |
| | Medicare | 34.53 |
| | Medicaid | 7.23 |
| Language | English | 90.84 |
| | Other | 9.16 |
| Admit Type | Emergency | 56.95 |
| | Other | 41.60 |
| | Elective | 1.45 |

### F.7 FEATURE CORRELATION MATRIX FOR ELECTRONIC HEALTH RECORD

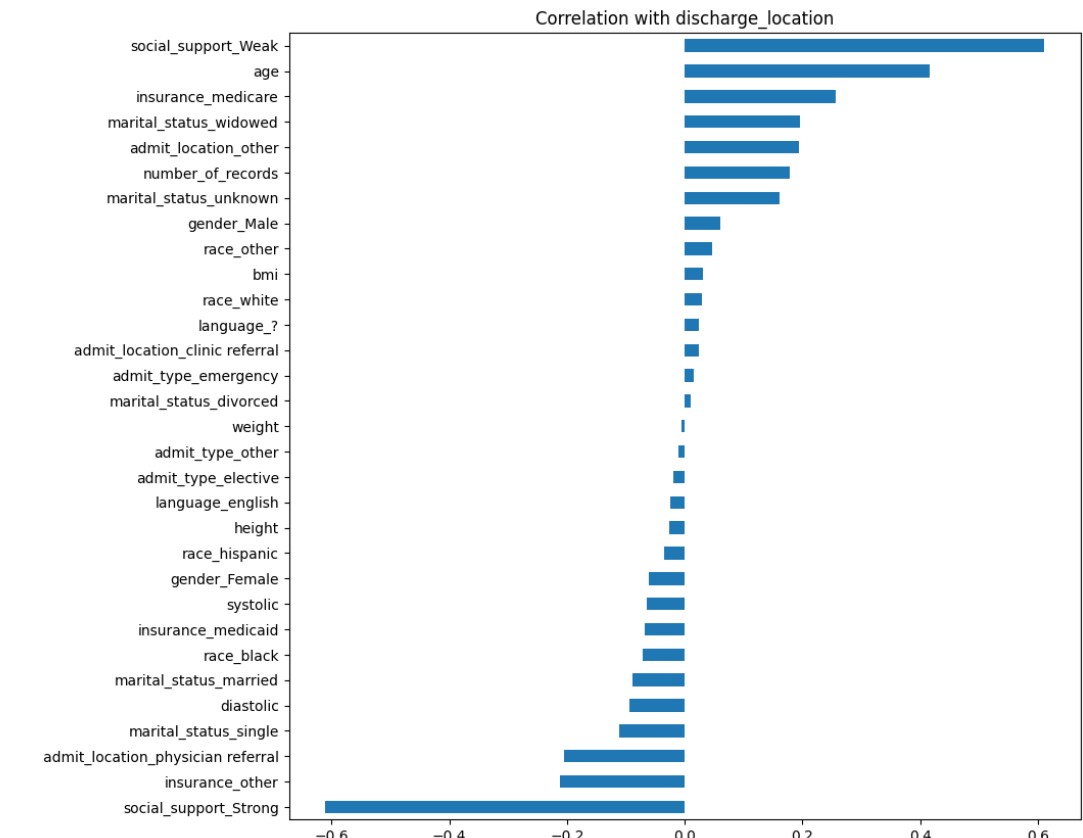

Figure 12: Correlation Matrix of features for Discharge Location Data

