# OpenReview forum: "Latent Feature Mining for Predictive Model Enhancement with Large Language Models"
_ICLR.cc/2025/Conference — Submitted to ICLR 2025_

### Official Review · Reviewer_JQW7 · 2024-10-30

**Soundness:** 3
**Presentation:** 3
**Contribution:** 3
**Rating:** 6
**Confidence:** 4

**Summary:**

This paper presents FLAME (Faithful Latent FeAture Mining for Predictive Model Enhancement), a novel framework that leverages large language models (LLMs) to extract latent features from limited observed data to enhance predictive modeling. The framework addresses challenges in domains where data collection is constrained by ethical or practical limitations. The authors validate their approach through two case studies: criminal justice system and healthcare, demonstrating significant improvements in downstream prediction tasks.

**Strengths:**

- Novel Framework: FLAME presents an innovative approach to latent feature mining by leveraging LLMs' reasoning capabilities, distinguishing it from traditional methods.
- Practical Significance: The framework addresses real-world challenges in domains where data collection is limited by ethical constraints. The case studies demonstrate clear practical value.

**Weaknesses:**

- Scalability Concerns: The framework requires domain expertise for Step 1 (formulating baseline rationales). While this leads to better results, it might limit scalability to new domains.

**Questions:**

1. Could the authors provide more details about the computational requirements and runtime comparisons between FLAME and traditional approaches?
2. How might the framework perform in domains where expert knowledge is less readily available or more ambiguous?
3. Could the authors elaborate on how the framework might be adapted for streaming data or online learning scenarios?

---

> ### Author Response · Authors · 2024-11-24
> **Response to Reviewer  JQW7**
>
> We sincerely thank the reviewer for their time and effort. We greatly appreciate Reviewer JQW7's **recognition of the novelty of our framework and its practical value**. Here are our point-by-point responses to the questions raised by the reviewer:
>
> 1. **Concern on scalability to adapt to new domains.**
>
> Reviewer JQW7 concerned about the requirement for domain expertise in step 1 of our proposed framework, FLAME, will limit the scalability to new domains. From our experience in the two domains (criminal justice and healthcare), **the workload for domain experts are not unreasonable because the tasks do not involve technical programming**. Instead, experts contribute by writing rationales based on their professional experience, a process that is both **intuitive** and **efficient** for those familiar with the domain. As noted in the paper, the inclusion of domain expertise and contextual information leads to huge improvements in both interpretability and performance. We believe this trade-off is justified, as the required human efforts and tailoring to new domains is worthwhile compared to the significant value it adds to the prediction.
>
> 2. **Comparison of the computational requirements between FLAME and traditional approaches.**
>
> We have elaborate the computational requirements of FLAME in Appendix B. Open-source LLMs (such as LLaMA) indeed require more computational resources ( such as GPU) than the traditional approach, however, advancements in black-box pre-trained LLMs have made many models (such as GPT4) accessible through simple API calls. This significantly reduces the barrier to adoption and makes FLAME more practical for a wider range of users and applications.
>
> 3. **How to adapt FLAME in domains when expert knowledge is ambiguous?**
>
> Our framework is designed for scenarios where expert knowledge is available and meaningful. In cases where expert knowledge is ambiguous or unavailable, FLAME may not be the most suitable approach, as its strength lies in leveraging clear and domain-specific insights for latent feature extraction and reasoning.
>
> 4. **Can FLAME be adapted for online learning scenarios, such as streaming data?**
>
> FLAME is not specifically designed for online learning or streaming data scenarios. Its current focus is on leveraging static datasets (batched processing), where domain expertise can be effectively applied. Extending FLAME to handle streaming data would require significant modifications to its structure and workflow, which is beyond the motivation and scope of our current framework, but could be an interesting future research direction.
>
> ----
>
> We sincerely thank you for your feedback and comments, and hope this response addresses your question. We are looking forward to further discussion. If our responses have addressed your concerns, we kindly request a reconsideration of the rating score. Thank you again for your valuable input!

---

### Official Review · Reviewer_U2yA · 2024-10-31

**Soundness:** 2
**Presentation:** 3
**Contribution:** 3
**Rating:** 5
**Confidence:** 4

**Summary:**

This paper proposes a framework that leverages LLMs to augment observed features with latent features and enhance the predictive power of ML models in downstream tasks. Two situations about criminal justice system and healthcare validate the framework.

**Strengths:**

1. This work introduces human knowledge into existing domain text classification tasks, such as law and medicine, which can effectively improve the accuracy of downstream tasks and provide some interpretations.
2. This work provides a new idea for how to maximize prediction performance when data is limited, and it may inspire more related work.

**Weaknesses:**

1. The narrative angle of this work is very novel, but in my opinion, it essentially uses the human knowledge of LLM and CoT to complete the classification task of domain text， which lacks technical innovation.
2. The necessity of fine-tuning LLM need to be discussed more thoroughly. Fine-tuning LLM will cause it to lose most of its knowledge and multi-turn dialogue capabilities, and it will consume a lot of computing resources. A more reasonable approach is to train BERT-like models to classify the analysis generated by LLM. In particular, although the authors tried to fine-tune the LLM, it still failed to achieve better performance than the machine learning methods.
3. The necessity of components of the proposed method is unclear (for example, the necessity of reasoning). Being more specifically, the submission is lack of comparable baseline methods. The authors only verified that their method can achieve better performance than not using this method, but they do not compare with other methods that also utilize LLM knowledge for domain classification tasks. A simple baseline approach could be to use the LLM to directly generate an analysis of the question, and then use a text model (such as BERT) to classify both the question and the LLM analysis.

**Questions:**

1. I wonder if the author has considered the inconsistency of the content generated by LLM? There have been many studies [1] that have found that multiple prompt LLMs can generate inconsistent results, which may have a significant impact on the classification results.
- [1] Statistical knowledge assessment for large language models. NeurIPS 2023.

---

> ### Author Response · Authors · 2024-11-24
> **Response to Reviewer U2yA**
>
> We sincerely thank the reviewer for their time and thoughtful feedback. Our innovation is the proposal of transforming latent feature mining task into propositional logic reasoning task, which is provided to be working in two different domains. Here are our response to questions and concerns from reviewer U2yA:
>
>  1. **What is the necessity of components of the proposed method ?**
>
> We conducted the ablation study to demonstrate that each component of our method is important. The results are shown in Appendix D.
>
> 2.  **Why is fine-tuning needed?**
>
> The fine-tuning process **incorporates feedback loops with domain experts to adjust and correct the LLM’s reasoning pathways**, ensuring that the latent features inferred, such as the need for various programs, are aligned with nuanced, causal relationships rather than superficial statistical correlations (see our response to Reviewer cb5e). We also elaborate the importance of finetune in the main paper and in the ablation study (Please see appendix D for more details).
>
>
> 3. **Why is reasoning needed?**
>
> The reviewer proposes a simple baseline to use the LLM to directly generate an analysis of the question, and then use a text model (such as BERT) to classify both the question and the LLM analysis. Thank the reviewer for this interesting suggestion. While we appreciate this suggestion, we would like to clarify the motivation behind our work and how it differs: Our primary goal is to leverage LLMs to generate human-like reasoning processes, especially in **low-resource settings** where there is insufficient data to train text models for classification. The design of a step-by-step reasoning process offers the following advantages:
>
> (1).   **Challenges with Direct Long-Content Analysis**: As shown in Appendix D, Figure 10(a), the performance of LLMs in zero-shot and few-shot scenarios is poor and unstable, highlighting the necessity of fine-tuning to guide LLMs toward human-like reasoning. The proposed baseline would require substantial effort from human annotators to produce the necessary training data for fine-tuning, which is infeasible in low-resource settings.
>
> (2).  **Simplification through Step-by-Step Reasoning**: Our reasoning approach decomposes complex questions into multiple steps, making it significantly easier than directly generating a long-content analysis. Transparency and Interpretability: The structured reasoning process provides a clear, step-by-step logical flow, which is crucial for maintaining the transparency and interpretability of the COT process. It also helps retaining the causal relationship as we respond to Reviewer cb5e. This transparency and interpretability are especially desirable in settings where human oversight and understanding of AI decisions are critical.
>
>
> 4. **Concern on inconsistent results of LLMs from multiple prompts.**
>
> Thank you to the reviewer for raising the concern about inconsistent results of LLMs across multiple prompts. We acknowledge that inconsistency poses a potential risk to the robustness of LLMs. **This has been an important consideration in our implementation, and we have taken steps to guide LLMs to respond more robustly and reliably**: First, we fine-tune the model to ensure it generates text strictly adhering to the predefined format. This reduces the likelihood of variability in responses caused by different formatting. Second, we carefully design and test prompts to minimize ambiguity and guide the model toward consistent interpretations and outputs.
>
> For further mitigation, we can aggregate results from multiple prompts and apply filtering or voting mechanisms to enhance robustness and reduce outlier responses.
>
> ----
>
> We sincerely thank you for your feedback and comments, and hope this response addresses your question. We are looking forward to further discussion. If our responses have addressed your concerns, we kindly request a reconsideration of the rating score. Thank you again for your valuable input!

---

> > ### Comment · Reviewer_U2yA · 2024-11-25
> >
> > Thanks for your reply, which partially addresses my issues. Using LLM does make sense in terms of interpretability. But I think the response in 3(1) is somewhat self-contradictory. Existing text summarization models (such as BERT, or the newer GTE, etc.) are pre-trained on a large amount of data, so I don’t think fine-tuning a text summarization model requires more data than fine-tuning an LLM. In addition, without considering interpretability, I guess using MLP to classify the embeddings extracted by the text summarization model may be more effective. I hope the author can add this experiment.

---

> > > ### Author Response · Authors · 2024-11-26
> > > **Followup discussion**
> > >
> > > Thank you for your response. We are pleased that our previous explanations have addressed some of your concerns. Let us now address your remaining points:
> > >
> > > 1.  **Clarification on 3(1).**
> > >
> > > As mentioned in our paper, there is a key challenge in real-world applications: the absence of ground truth for latent features. To address this challenge, our approach augments the training data to finetune LLMs in self-instructed fashion. More specifically, human experts craft a handful of rationales, then we leverage the few-shot and instruction-following ability of LLMs to augment the size of these rationales and then use them as the training data for fine-tuning. During this process, we incorporate the automated evaluation metrics to ensure the quality of generated CoT, and involve human-in-the-loop to verify sampled CoT.
> > >
> > > As far as we understand, the baseline proposed by the reviewer is to use a text classification model (e.g., BERT) to process both the original question and the LLM-generated analysis as input features. This classification model would then predict the desired outcome (e.g., program needs) based on this combined information. Training this classification model would require the groundtruth label of each LLMs’ analysis, which can only be achieved by human annotation for higher quality. Hence, we believe that preparing training data for this baseline requires more human effort than our proposed method to prepare training data for fine-tuned LLMs.
> > >
> > >
> > > 2.  **Implemented baseline as reviewer suggested**.
> > >
> > > Following reviewer suggestion, we implemented a baseline approach using an MLP classifier on embeddings from a text summarization model, and repeat the risk level prediction experiment (Section 5.2) : We use few-shot to generate a profile that contains all information related to the client ( See Appendix C, figure 6 for more detail on prompt template ), then we extract the embedding from the encoder of Pegasus (strong model pre trained on abstractive summarization ) [1]. These embeddings serve as input to an MLP classifier for risk level prediction. Due to time and computational constraints, we evaluated on a balanced validation set of 50 samples per class, and this baseline achieved only 52% accuracy on the three-class classification task, while our proposed approach is able to reach over 75% accuracy. This experiment’s results are added in Appendix D of our revised PDF (uploaded today).
> > >
> > >
> > > 3.  **Ablation experiment for the importance of “reasoning”**.
> > >
> > > To further validate the importance of the “reasoning” step, we conducted an ablation study examining **how different reasoning strategies affect risk level prediction accurac**y (the experiment setting in Section 6.1). This is in Appendix D of our paper. Our investigation focused on two key parameters: the number of reasoning steps and the length of reasoning sentences (n). These parameters were systematically varied to understand their impact on prediction performance. Figure 10 (b) in the appendix shows the impact, while we list the numeric values below in the table:
> > >
> > >
> > > | Number of Reasoning Steps | \( n = 1 \) | \( n = 2 \) |
> > > |---------------------------|-------------|-------------|
> > > | 4                         | 0.50        | 0.62        |
> > > | 5                         | 0.55        | 0.64        |
> > > | 6                         | 0.60        | 0.66        |
> > > | 7                         | 0.62        | 0.68        |
> > > | 8                         | 0.65        | 0.70        |
> > > | 9                         | 0.68        | 0.72        |
> > > | 10                        | 0.70        | 0.75        |
> > >
> > >
> > > Across most sentence lengths, **increasing the number of reasoning steps consistently improved performance**. The accuracy rose from 0.50 to 0.70 for n=1, and from 0.62 to 0.75 for n=2. This suggests that more granular reasoning leads to better predictions. The best performance (0.75 accuracy) was achieved with 10 reasoning steps and a sentence length of 2, significantly outperforming the single-turn zero-shot baseline of 0.55.
> > >
> > > ----
> > >
> > > We sincerely thank you for your feedback and comments, and hope this response addresses your question. **We are looking forward to further discussion. If our responses have addressed your concerns, we kindly request a reconsideration of the rating score**. Thank you again for your valuable input!
> > >
> > > ----
> > >
> > > [1] PEGASUS: Pre-training with Extracted Gap-sentences for Abstractive Summarization. [https://huggingface.co/google/pegasus-cnn_dailymail](https://huggingface.co/google/pegasus-cnn_dailymail)

---

### Official Review · Reviewer_HEk5 · 2024-11-04

**Soundness:** 2
**Presentation:** 2
**Contribution:** 1
**Rating:** 3
**Confidence:** 3

**Summary:**

This paper proposes an approach to the creation of latent features capturing aspects of data not directly observed. These aspects, termed latent features, are modeled from a given dataset using an LLM interaction framework involving the generation of rationales produced using principles from propositional logic and prompting techniques. The estimated latent features are then fed into a predictive model, where there is evidence of improved predictive performance in various case studies involving health care and criminal justice.

**Strengths:**

There are aspects of the paper I perceive to be strengths given my understanding of it.

The translation step of unstructured text into predicates for LLM processing is clever. It bears some similarities to the DataTuner system but seems like a useful approach.

I appreciate that the paper includes as clear baseline against which evaluations are performed.

The case studies are drawn from diverse fields and seem to have practical importance.

**Weaknesses:**

There are aspects of the paper I perceive to be weaknesses given my understanding of it.

For example, the conceptualization of "latent feature" seems to be a bit contradictory at times. For example, in the DAG on p. 3 implies that "latent features" are to be mediators of X and its effect on Y. Later on (for example, in Figure 3, simple correlation (with bi-directional arrow) is postulated between latent and observed features). In the examples of (e.g.) Figure 2, there also seems to be potentially ambiguous causal ordering. It's unclear to me what the implications of this would be, but some of these questions regarding definition make the paper harder to interpret.

Also, in a way, traditional neural network models, with many hidden layers, are more or less entirely about modeling latent numerical variables. The approach here seems to be a specific way of generating human-interpretable latent variables from one neueral network model (in this case, an LLM), for improving performance on the original neural network task. It wasn't entirely clear to me why interpretability was important in these tasks...

More importantly, given my understanding of the paper, two major kinds of items are compared. First, input features X are used to predict Y. Second, input features are augmented with the latent values processed from X, and used to predict Y. The paper finds improved performance of the latter approach. However, this seems to me to be far from an apples-to-apples comparision, as in the first approach, (I believe) a very relatively neural model is fit (I couldn't find the exact parameter count). In the second approach, a muilti-billion parameter is used to generate features that the smaller neural model then uses. On this account, model scale alone makes interpretation of the results difficult.

I believe a stronger evaluation would be to employ equally sized neural models in the two arms of the experiment (forgive me if this is being done and I misunderstood the approach taken, in particular, in Experiment 1). I would be more convinced with some combination of the following. (1) Compare against a neural embedding of the structured data (perhaps using the general approach taken to render the X's amendable to LLM processing). This way, we would have a model with roughly the same number of parameters being used across arms, and we could better understand whether it was the specific way of generating Z's, or the general use of large-scale neural/LLM models that was generating improved performance over the baseline methods. (2) I would also be more convinced if performance were improved over simpler prompting-based strategies to generating outcome predictions (as this would be another way to employ the multi-billions of parameters to this task, rendering the comparision with the specific strategy used here more rigorious). The two approaches just described would involve similar levels of compute used for the other experiments already run, but would help put the results in context.

In general, a much simpler approach seems to be to use the LLM to generate direct predictions of the outcome, where it will form many latent representations internally. The prompts outlined in the Appendix do this for predicting the Z's (where here, there is ground truth data for the Z's, so these are in some of the applications observed nodes). The interpretability angle is important but can be difficult to define or defend; I don't seem to see much evidence directed towards this claim in the results section of the paper.

**Questions:**

What are the number of trainable parameters used in the MLPs trained here?

**Details Of Ethics Concerns:**

No major ethical concerns.

---

> ### Author Response · Authors · 2024-11-24
> **Response to Reviewer HEk5 [1/2]**
>
> We sincerely thank the reviewer for their time and thoughtful feedback. We particularly appreciate reviewer HEk5’s comment that our approach, **FLAME, is both clever and demonstrates practical importance**. Here are our response to questions and concerns from reviewer HEk5:
>
> 1. **Clarification on the definition of latent feature.**
>
> Thank the reviewer HEk5 for pointing out the inconsistency in the visual illustration of the latent features. The latent feature (Z) is correlated with both input feature (X), and prediction target (Y). We will revise the DAG figure to more clearly illustrate the mutual correlation between X and Z during the final revision. Thank the reviewer HEk5 for this valuable suggestion.
>
>   2. **Why is interpretability important for prediction?**
>
> Interpretability is crucial for prediction, especially in high-stakes areas like healthcare and criminal justice, because it ensures we understand the role and impact of each feature in the decision-making process. Moreover, our FLAME approach produces latent features that are directly interpretable, which allow users to see how these features are deducted from step-by-step rationale. **This process builds trust, enables accountability, and helps identify potential biases or errors during the rationing process.** Without interpretability, using predictions in critical scenarios could lead to unintended consequences, as we wouldn't be able to justify or validate the outcomes.
>
> 3. **Model scale difference (million-parameters LLMs V.S. smaller neural networks) makes evaluation difficult.**
>
> The reviewer may have misunderstood the key comparison, which pertains to the experiment in Section 6.2 rather than Section 6.1. Specifically, in Section 6.2, we extract a single latent feature using FLAME and evaluate its impact on prediction performance by comparing models of comparable size (LR, MLP, and GBT) with and without the latent feature. Your question is more relevant to the experiment discussed in Section 6.1, which we address below.
>
> 4. **It's not a fair comparison in experiment 1, since the LLMs are extremely larger than neural networks.**
>
> We acknowledge the misunderstanding regarding the purpose of Experiment 1. The experiment was designed to (i) **validate the ability of LLMs to mimic human thinking processes** for extracting latent features and (ii) **verify that the extracted latent features are accurate, rather than directly comparing LLMs with traditional ML models**. Specifically, we simulated a scenario as if the ground truth label is missing and aimed to evaluate LLMs' performance in following instructions and accurately extracting latent features based on human expertise and established risk scoring rules.
>
> To prevent similar confusion for other readers, we will move the plot and chart of the Experiment 1 results to the appendix, removing the comparison with the ML models in the plot and charts. In the main paper, we will shift the primary focus to the remaining experiments, where LLMs are not directly used as predictive models. This restructuring should provide better clarity and emphasis on the core contributions of our work.
>
> Moreover, we want to mention that advancements in black-box pre-trained LLMs have made many models (such as GPT4) accessible through simple API calls. This significantly reduces the barrier to adoption and makes FLAME more practical for a wider range of users and applications.

---

> > ### Comment · Reviewer_HEk5 · 2024-11-25
> > **Thanks for these clarifications**
> >
> > Thanks to the authors for their clarifications here, especially regarding section 6.
> >
> > To avoid the same kind of confusion I had, It may help to add some explicit connections between different parts of the results section. For example, from what I understand, Figure 4 corresponds to Step 4 in Section 5.1. I wonder if there are ways to help the reader understand these results in context. For example, the methods section is initially introduced using case studies as the structuring principle, but in the discussion of results, there is a less clear mapping of figures/tables to the cases introduced earlier (although of course there are some connections).
> >
> > The suggested changes sound reasonable to me. Another suggestion would be to focus on one of the case studies, leaving the other for the appendix.

---

> > > ### Author Response · Authors · 2024-11-26
> > >
> > > We sincerely appreciate your feedback and constructive suggestions.
> > >
> > > Based on your suggestion, we have updated the DAG figure to ensure consistency in the definition of latent features. The change is reflected in the updated PDF. Due to the limited time, we will revise the manuscript to improve clarity and structure in the camera ready version. Specifically, we will rearrange the sections to establish more explicit connections between the results and the methods. For example, we will ensure that key figures, such as Figure 4, are explicitly mapped to corresponding steps in Section 5.1, as you noted. To avoid potential confusion, we will focus on the criminal justice case study in the main text and move the majority of the details of the healthcare case study to the appendix (while leaving some highlights in the paper so that readers would appreciate that the framework works on different domains). This will streamline the narrative and help readers follow the results in context.
> > >
> > > We hope these revisions will address your concerns and improve the manuscript's clarity. Given these changes, we kindly request a reconsideration on our merits if you feel the updated manuscript aligns better with your expectations. Thank you again for your valuable input! We are looking forward to further discussion.

---

> ### Author Response · Authors · 2024-11-24
> **Response to Reviewer HEk5 [2/2]**
>
> 5. **Can we use LLMs as a predictive model to generate the prediction of outcome directly?**
>
> Recent works suggest that LLMs still cannot surpass typical traditional ML models such as XGBoost, SVM, Transformer and RNN [1]. The downstream task (e.g., outcome prediction) usually relies on multiple different variables, the training data, and domain-specific relationship, which makes it harder for general LLMs to do the prediction. FLAME is leveraging the domain expertise to infer the latent features and then enhance the downstream tasks. In fact, we have conducted an ablation experiment to test LLMs’ ability in zero-shot, which is equivalent to using LLMs to directly generate predictions of the outcomes as you suggested. As result shown in Appendix D Figure 10 (a), zero-shot demonstrates poor performance in inference. This suggests that direct outcome prediction is challenging for LLMs .
>
>
> 6. **What are the number of trainable parameters used in the MLPs trained here?**
>
>
> The MLP trained has the input size of  30, three hidden layers (50, 60, and 20 neurons), and an output layer of 10 neurons has a total of 6,040 trainable parameters. These are calculated as follows: the connection from the input to the first hidden layer contributes `(30 × 50) + 50 = 1,550` parameters, from the first to the second hidden layer contributes `(50 × 60) + 60 = 3,060`, from the second to the third hidden layer contributes `(60 × 20) + 20 = 1,220`, and from the third hidden layer to the output layer contributes `(20 × 10) + 10 = 210`. Summing these gives a total parameter count of 6,040.
>
> ----
>
> We sincerely thank you for your feedback and comments, and hope this response addresses your question. We are looking forward to further discussion. If our responses have addressed your concerns, we kindly request a reconsideration of the rating score. Thank you again for your valuable input!
>
> ----
>
> [1] Chen, C., Yu, J., Chen, S., Liu, C., Wan, Z., Bitterman, D., ... & Shu, K. (2024). ClinicalBench: Can LLMs Beat Traditional ML Models in Clinical Prediction?. arXiv preprint arXiv:2411.06469.

---

### Official Review · Reviewer_cb5e · 2024-11-05

**Soundness:** 2
**Presentation:** 3
**Contribution:** 2
**Rating:** 3
**Confidence:** 4

**Summary:**

The paper  proposes FLAME, a framework that uses LLMs for inferring latent features through text-based propositional reasoning. Its goal is to improve predictive models in data-limited domains. FLAME reframes latent feature extraction as a reasoning task, converting observed features into logical propositions that LLMs process to derive latent features. The framework formulates rationales, generates synthetic data, fine-tunes the LLM, and then infers latent features. In case studies from criminal justice and healthcare, FLAME outperforms traditional ML models.

**Strengths:**

The paper presents FLAME, a framework for enhancing predictive models by inferring latent features through large language models (LLMs) using text-based reasoning. The approach is novel in its application of LLMs to simulate expert-like reasoning in data-limited contexts, aiming to improve predictions in fields like criminal justice and healthcare. The quality of the work is solidly supported by experiments, though some reliance on domain-specific inputs could limit broader applicability. The framework is generally well-explained, although some technical descriptions, especially around the logical reasoning structure, could be clearer. Overall, FLAME offers a creative step towards improving model interpretability and performance in constrained data settings.

**Weaknesses:**

1. In safety-critical domains like criminal justice and healthcare, understanding causal relationships is what we actually need, including for predictive modeling. At the very least, to understand if the approach genuinely learns meaningful patterns or simply exploits spurious correlations, the models need to be tested on OOD data (as is common in this of literature within ML).

2. The ability of the LLM to uncover the "true" latent variables via prompting doesn't seem convincing. Maybe through latent variable modeling on the model's representation space we could learn something better. Alternatively, researchers working on these problems often use discovery methods like topic models or ideal-points models. More broadly, the social science literature is very aware of the limitations in using black-box ML for variable discovery.

3. Simply introducing LLMs into safety-critical domains without transparent guarantees does not align with how researchers in these fields approach variable discovery. Without understanding or validating what the LLM captures, this approach risks missing the mark on reliability and ethical standards essential for high-stakes decisions.

**Questions:**

1. Given that causal relationships are critical in domains like criminal justice and healthcare, how does the framework ensure that the LLM-generated latent features capture causally relevant information rather than exploiting correlations?

2. The LLM-based prompting for latent feature discovery seems less principled compared to structured methods like topic models or ideal-point models. Could you explain why this approach was chosen and provide any empirical or theoretical comparison to these established techniques?

3. Since the framework is intended for use in safety-critical domains, interpretability and reliability are essential. Can you provide more detail on how they validate the inferred latent features, especially when ground-truth labels for these features are not available?

4. The paper claims domain flexibility, but it also seems FLAME relies on significant customization, especially in formulating baseline rationales. Could you clarify the extent of domain-specific adaptation required and discuss any general principles or guidelines for adapting FLAME to new domains?

5. Given the sensitive nature of the target domains, could the authors expand on their ethical considerations? Specifically, are there safeguards to prevent biases from the LLM from impacting inferred latent features?

---

> ### Author Response · Authors · 2024-11-24
> **Response to Reviewer cb5e [1/2]**
>
> We sincerely thank the reviewer for their time and thoughtful feedback. We are particularly grateful to reviewer cb5e for acknowledging that "**the quality of the work is solidly supported by experiments**" and recognizing that our model, FLAME, represents a **creative and significant step** towards enhancing model interpretability and performance in constrained data settings. Below, we provide detailed responses to the reviewer’s questions and concerns:
>
>
> ---
>
> 1. **How to ensure that the method learns latent information that is causally predictive of the outcome ?**
>
> Our approach leverages expert knowledge to mine latent features. In particular, the human crafted rationales have a justified causal link to the outcomes based on the human expert knowledge, as well as known relationships in the literature. For example, in the criminal justice domain, social-economic status is known to have a causal effect on probation program outcomes[1,2], and this is verified by our community partner (human experts) when developing the basic rationales. The design of our approach places significant emphasis on the integration of human expertise and domain knowledge: We decompose the complex rationales to a series of simple rationales. In each rationale, we have clear instructions to guide the reasoning, and finetune on human crafted rationales to make LLM’s reasoning aligned with human instruction. In other words, **we are not letting LLMs randomly mine features that may have just correlations with the outcome – we are using a rigorous approach to allow LLMs mimic the human reasoning process that follows a causal relationship**. Since the human crafted rationales have the casual relationship, the fine-tuned LLMs that can accurately mimic the reasoning process should also be able to identify latent features that are causally predictive of the outcome.
>
> To further validate this, we conduct two additional human evaluations on the LLMs generated rationales.
>
>  In the first evaluation, we randomly sampled LLMs rationales used in experiment 1 and experiment 2 of the criminal justice case study (20 rationales in total). We invited three experts in criminal justice ( denoted as e1, e2, e3) , and two students who don't have any prior knowledge on this work (denoted as u1, u2) to annotate each rationale. We asked following two questions: (1) "*The correctness of the reasoning process: Is the LLM's reasoning consistent with how humans causal reasoning?*" (2)  "*The correctness of final conclusion: Would you agree with the reasoning step as a valid human response to the task?*"
>
> The results are shown below:
> |                                                   | e1   | e2   | e3  | u1   | u2   |
> |---------------------------------------------------|------|------|-----|------|------|
> | Percentage of rationales with correct reasoning   | 80%  | 100% | 75% | 95%  | 100% |
> | Percentage of rationales with correct conclusion  | 100% | 100% | 90% | 100% | 100% |
>
> These results showcase the strong ability of LLMs to replicate human crafted rationales, and make accurate decisions.
>
> The second evaluation is designed to test whether humans can distinguish the reasoning process generated from human or LLMs. We mix the sampled rationales with 20 more human crafted rationales, and we asked the five human annotators with this question: "*Do you think these reasoning steps were generated by a Human or an LLM?*"
>
> The result shows that, on average, 75% of human crafted rationales are wrongly annotated as LLMs generated, and 60% of LLMs generated rationales are wrongly annotated as human crafted. We believe these results further validate the quality of generated rationales by LLMs.
>
> In conclusion, the two additional human evaluations demonstrate that **LLMs is able to accurately mimic the human crafted rationales (that are justified to capture the underlying causal relationship), so the latent features mined by LLMs should also be causally predictive of the outcome.**

---

> ### Author Response · Authors · 2024-11-24
> **Response to Reviewer cb5e [2/2]**
>
> 2. **What is the advantage of FLAME over structured methods like topic models or ideal-point models ?**
>
> Our approach is fundamentally different from topic models or ideal-point models. These traditional methods often rely on predefined parametric assumptions, such as specific distributions (e.g., Dirichlet for LDA or normality for ideal-point models) and primarily work with structured or semi-structured datasets, such as text corpora for topic models or specific preference/ranking data for ideal-point models. Also their output is not directly interpretable – Latent features in topic models are typically abstract clusters or distributions over words that lack direct interpretability unless post-processed or annotated by humans.
>
> Ideal-point models produce latent traits tied to a specific metric (e.g., ideology or preference score) but do not capture richer, contextualized relationships beyond the given data. In contrast, FLAME is designed to process both structured and unstructured data, making them highly versatile. Our approach can incorporate contextual, domain-specific information expressed in natural language, which allows for reasoning beyond rigid data structures.This makes it more flexible and better suited for handling complex, diverse datasets. Moreover, the latent features inferred by LLMs are generated through reasoning and can capture nuanced relationships (including causal relationship as we addressed in your first question), domain-specific knowledge and other contextual information. The mined latent features are directly interpretable because they can be aligned with natural language rationales, allowing for greater transparency in how they are derived. As a result, **FLAME can adapt more effectively to real-world tasks and uncover insights that traditional models might miss, especially for domains where contextual, domain-specific information is critical, such as healthcare or criminal justice.** The interpretability is particularly useful for high-stakes decisions.
>
>
>
> 3. **How to validate the inferred latent features when there is a lack of ground truth?**
>
> We have human validation before the finetune as described in the paper, and experiment 1 in the criminal justice case study shows that the inferred latent feature is trustworthy. Moreover, we have conducted additional human evaluation for the extracted latent features (see our response to your first question).
>
> 4. **Can you clarify the extent of domain-specific adaptation required?**
>
> We need input from domain expertise to help craft rationales in natural language (Step 1). The rest of the steps can be easily adapted to new domains. From our experience in the criminal justice and healthcare domain, this step definitely requires human experts but the actual workload is not heavy for experts since they don’t need to do the programming-required tasks. As discussed in the last section of our paper, we believe this effort is worth it as it allows incorporation of useful contextual information and expert knowledge.
>
> 5. **Are there any guidelines or principles required for adapting the FLAME to a new domain?**
>
> The general guidelines to adapt FLAME to new areas is to decompose the complex rationales to a group of simple rationales, which could be best understanded and learned by LLMs.
>
> 6. **Are there safeguards to prevent biases from the LLM from impacting inferred latent features?**
>
>
> We have human validation before the finetune. In addition, we further conducted bias analysis (as shown in the Appendix D) to ensure there’s no bias. We are also conducting human evaluation.
>
> ----
>
> We sincerely thank you for your feedback and comments, and hope this response addresses your question. We are looking forward to further discussion. If our responses have addressed your concerns, we kindly request a reconsideration of the rating score. Thank you again for your valuable input!
>
>
> ----
>
> [1] Steinmetz, K. F., & Henderson, H. (2015). Inequality on probation: An examination of differential probation outcomes. In Journal of Ethnicity in Criminal Justice (Vol. 14, Issue 1, pp. 1–20). Informa UK Limited. [https://doi.org/10.1080/15377938.2015.1030527](https://doi.org/10.1080/15377938.2015.1030527)
>
> [2] Administrator. (2024, February 7). Socioeconomic status and criminal justice outcomes - iresearchnet. Criminal Justice. https://criminal-justice.iresearchnet.com/criminal-justice-process/racial-and-socioeconomic-disparities/socioeconomic-status-and-criminal-justice-outcomes/

---

> > ### Author Response · Authors · 2024-12-02
> >
> > Dear Reviewer cb5e,
> >
> > As the discussion period is nearing its end, we wanted to ensure you've had a chance to review our detailed response addressing your key concerns:
> >
> > - Casual Relationship Concern: We provide detailed explanation, and conducted additional human evaluation to further validate.
> > - Comparison with other models: We elaborate the difference.
> > - Clarification of the extent of domain-specific adaptation required: We provide detailed clarification, and guidelines for other researchers.
> >
> > We are looking forward to further discussion. If our responses have addressed your concerns, we kindly request a reconsideration of the rating score. Thank you for your consideration.
> >
> > Thank you again for your valuable input!

---

### Meta-Review · Area_Chair_Hmiu · 2024-12-17

**Metareview:**

**Meta-Review: Reject**

This paper proposes FLAME, a framework that uses LLMs for inferring latent features through text-based propositional reasoning, aiming to improve predictive performance in data-limited domains. While the concept of leveraging LLMs for latent feature discovery is novel and has practical applications in domains like healthcare and criminal justice, the paper suffers from several critical weaknesses. First, the evaluation lacks rigor, as comparisons between models are not apples-to-apples: the baseline models are relatively simple, while FLAME benefits from the latent representations of a multi-billion parameter LLM, making it unclear whether the gains are due to scale rather than the proposed reasoning framework. Second, the necessity and innovation of key components, such as fine-tuning LLMs and reasoning steps, remain underexplored; alternative baselines (e.g., direct LLM predictions or BERT-based embeddings) are missing, which weakens the claims. Third, the interpretability motivation is not convincingly justified or demonstrated, as no evidence is provided that the latent features learned through FLAME are meaningful or reliable, particularly in safety-critical domains. Finally, the lack of robustness testing on out-of-distribution (OOD) data and ambiguous causal definitions of latent variables raise further concerns about the validity and applicability of the method. While the idea is promising, the technical and experimental shortcomings prevent the paper from making a strong contribution in its current form.

**Additional Comments On Reviewer Discussion:**

Reviewers provided great feedback on how to improve the paper, especially regarding the motivation, problem setting, and evaluation. While the authors tried to address these by providing more details and arguments, these are fundamental limitations of the work which cannot be addressed during the short period of time during rebuttal.

---

### Decision · Program_Chairs · 2025-01-22

Reject